# Increasing Model Capacity for Free: A Simple Strategy for Parameter Efficient Fine-tuning

**Haobo Song**[2,*]   **Hao Zhao**[2,*]   **Soumajit Majumder**[3]   **Tao Lin**[1,†]

haobo.song@epfl.ch;   hao.zhao@epfl.ch;
soumajit.majumder@huawei.com;   lintao@westlake.edu.cn
[1]Westlake University   [2]EPFL   [3]Huawei

## Abstract

Fine-tuning large pre-trained foundation models, such as the 175B GPT-3, have attracted more attention for downstream tasks recently. While parameter-efficient fine-tuning methods have been proposed and proven effective without retraining all model parameters, their performance is limited by the capacity of incremental modules, especially under constrained parameter budgets.

To overcome this challenge, we propose CAPABOOST, a simple yet effective strategy that enhances model capacity by leveraging low-rank updates through parallel weight modules in target layers. By applying static random masks to the shared weight matrix, CAPABOOST constructs a diverse set of weight matrices, effectively increasing the rank of incremental weights without adding parameters. Notably, our approach can be seamlessly integrated into various existing parameter-efficient fine-tuning methods. We extensively validate the efficacy of CAPABOOST through experiments on diverse downstream tasks, including natural language understanding, question answering, and image classification. Our results demonstrate significant improvements over baselines, without incurring additional computation or storage costs. Our code is available at https://github.com/LINs-lab/CapaBoost.

## 1 Introduction

In recent years, the prevailing training paradigm has revolved around pre-training models on large-scale datasets and subsequently fine-tuning them for diverse downstream tasks, yielding remarkable achievements. However, the increasing size of popular pre-trained models, such as LLaMA2 (Touvron et al., 2023) and GPT3 with a size of 175B (Floridi & Chiriatti, 2020), poses significant challenges for full-sized model fine-tuning. Memory and storage limitations restrict its practicality and applicability.

Parameter Efficient Fine-Tuning (PEFT) emerges as a compelling solution to address these challenges head-on. Unlike the resource-intensive nature of full-size fine-tuning, PEFT adopts a judicious approach by either fine-tuning a small subset of the original model's parameters or introducing a limited number of additional parameters during the fine-tuning process. This strategy effectively alleviates the memory and computation burdens, providing a cost-effective alternative that can match or surpass the performance of full-size fine-tuning. Prominent PEFT techniques employed today include Prompt Learning (Sun & Lai, 2020), Prefix-Tuning (Li & Liang, 2021), Adapters (Houlsby et al., 2019; Pfeiffer et al., 2021), and LoRA (Hu et al., 2022).

The core concept of these PEFT techniques lies in approximating a single layer's heavy-weight matrix $\mathbf{w} \in \mathbb{R}^{d_1 \times d_2}$ by two consecutive layers possessing much smaller inner dimension $r$ such as $\boldsymbol{B} \in \mathbb{R}^{d_1 \times r}, \boldsymbol{A} \in \mathbb{R}^{r \times d_2}$. This approach results in a significant reduction in parameter count (evident in Adapters and LoRA), achieving performance comparable to full-size fine-tuning while retaining only 1% of trainable parameters. However, the inner dimension cannot be arbitrarily small, as its impact on performance is substantial. For instance, in LoRA, the trainable parameter $\boldsymbol{BA}$ is constrained by rank$(\boldsymbol{BA}) \leq r$ (Hu et al., 2022), setting an upper bound on the model's capacity. These capacity constraints often lead to suboptimal performance when $r$ is small (He et al., 2021).

---

[*]Equal contribution.
[†]Corresponding author.

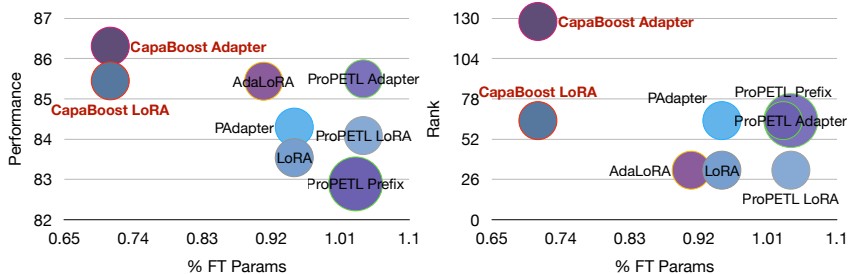

Figure 1: **Rank and Performance comparison among several PEFT methods in GLUE test with RoBERTa base model.** The left figure shows the performance and parameter numbers (shown as percentage to fully fine-tuning) of different methods and indicates CAPABOOST is the best. The right figure shows the rank and parameter numbers of several methods and CAPABOOST has the highest rank among similar PEFT methods.

The pursuit of an innovative method capable of reducing parameters while preserving a high model rank becomes paramount. To tackle this challenge, we present CAPABOOST, a versatile framework aimed at *increasing model capacity for free* in various PEFT approaches, e.g., Prefix-Tuning, Adapters, LoRA, and their variants. Through the integration of model pruning and weight sharing, CAPABOOST significantly amplifies model capacity while simultaneously reducing parameter count, all without inflating FLOP numbers. Leveraging the elevated rank and expanded model capacity, CAPABOOST PEFT methods demonstrate substantial performance improvements over their original counterparts, as demonstrated in Figure 1. Our contributions are summarized as follows:

- We introduce CAPABOOST, a plugin framework that seamlessly integrates with various PEFT methods, such as Adapters and LoRA. Through model pruning and weight-sharing, CAPABOOST enhances model rank without incurring additional costs.
- Extensive results unequivocally demonstrate that CAPABOOST outperforms all state-of-the-art baselines while significantly reducing parameter count and maintaining the same or fewer FLOPs.

## 2 RELATED WORK

We provide a compact summary here due to space issues. A complete discussion is in Appendix J.

**Parameter-Efficient Fine-tuning (PEFT).** In the realm of PEFT, the initial approach involves selecting a subset of the original model parameters for updating, such as top layers (Donahue et al., 2014), specific model layers (Gheini et al., 2021), or internal modules (Zaken et al., 2022). While these brute-force selection methods effectively reduce the number of trainable parameters, they often result in sub-optimal performance. Consequently, various scoring functions have been employed to assess the importance of each parameter (Sung et al., 2021; Ansell et al., 2022; Guo et al., 2021). However, these scoring functions are typically task-specific and involve additional computational overhead.

Another line of work proposes sharing the pre-trained network and incorporating task-specific trainable modules, significantly reducing storage costs (Zhao et al., 2020). Notably, HAdapter (Houlsby et al., 2019) introduces adapter modules after each feed-forward and attention layer, while PAdapter (Pfeiffer et al., 2021) further suggests using adapters after FFN and LayerNorm modules (Ba et al., 2016) for improved efficiency. Drawing inspiration from textual prompting methods (Sun & Lai, 2020; Liu et al., 2019; Jiang et al., 2020; Shin et al., 2020), Prefix-Tuning (Li & Liang, 2021) employs additional prefix vectors that are prepended to the keys and values in the attention layer. Building upon this, He et al. (2021) conducts a systematic exploration of existing advancements and proposes a diverse range of Adapter variants by incorporating design elements from Prefix-Tuning.

To alleviate the inference latency caused by incremental parameters, Hu et al. (2022) employ a bottle-neck structure with a low-rank constraint, enabling the merging of learned weights into the pre-trained network. However, these methods are limited by the prescribed rank values, which impose constraints on the maximum rank of trainable parameter matrices, thus limiting the model's capacity. In contrast, our work aims to overcome such capacity limitations and enhance parameter efficiency for fine-tuning.

**Low-rank Properties in Deep Neural Networks.** In the over-parameterized regime, it has been widely observed in various deep learning tasks that neural networks exhibit low-rank properties following training (Oymak et al., 2019). Building upon this insight, several works (Jaderberg et al., 2014; Sainath et al., 2013; Khodak et al., 2020) propose explicitly incorporating low-rank constraints during neural network training, leading to notable successes in CNNs. Following a similar approach,

LoRA (Hu et al., 2022) and subsequent studies (Dettmers et al., 2023; Zhang et al., 2022a; Chavan et al., 2023; Chen et al., 2023) adopt low-rank updates applied to a frozen pre-trained network for fine-tuning in downstream tasks.

While modern neural architectures like transformers have demonstrated (Aghajanyan et al., 2021; Wang et al., 2020) low-rank characteristics in their dimensions and representations, Bhojanapalli et al. (2020) highlight that the low-rank property of key and query projections in the multi-head attention modules becomes a bottleneck for transformer performance. Furthermore, experiments in Lialin et al. (2023) reveal that transformers with low-rank updates may under-perform that of full-rank baselines in some cases, especially when the fine-tuning weight rank is small.

## 2.1 Parameter Pruning for PEFT Methods

Weight-tied models, also referred to as weight-sharing or weight-tying models, represent a parameter-efficient neural network architecture where the same set of weights is shared across different layers or segments of the input (Dehghani et al., 2019; Dabre & Fujita, 2019; Xia et al., 2019; Lan et al., 2020; Li et al., 2021; Takase & Kiyono, 2021). Serving as the foundation for most implicit models, this architecture has garnered significant attention in recent years across a wide range of tasks (Wang et al., 2019; Liu et al., 2020; Yang et al., 2018; Lan et al., 2020; Takase & Kiyono, 2021; Zhang et al., 2020; Bender et al., 2020; Xie et al., 2021; Li et al., 2021). Pruning, an orthogonal yet extensively employed strategy, aims to enhance neural network efficiency by identifying and eliminating redundant parameters (Wang et al., 2022a; Frankle & Carbin, 2019; Lin et al., 2020; Su et al., 2020; Frankle et al., 2021; Wang et al., 2022b; He et al., 2022b). To the best of our knowledge, our idea of introducing various deterministic random masks to a parallel weight-tied model is novel.

The most relevant work to our method might be Bai et al. (2022), which extends the codebook idea and learns masks on top of a fixed random weight vector to represent diverse dense layers. Specifically, masks are utilized to select values from a random vector (i.e., codebook), resulting in layers with distinct structures. Simultaneously, Zeng et al. (2023) introduce ProPETL, a method that also incorporates shared weights but employs different masks to encode layer-specific and task-specific knowledge. Our work, in contrast to these approaches, learns shared weights with deterministic random binary masks, leading to stronger model capability.

AdaLoRA (Zhang et al., 2022a) offers an alternative approach to achieving parameter efficiency by dynamically allocating the parameter budget across different layers. This is accomplished through iterative pruning of singular values of incremental parameters based on an importance metric. However, it is worth noting that AdaLoRA entails additional computational overhead and necessitates a higher initial budget of trainable parameters, making it less suitable for low-resource scenarios.

## 3 CapaBoost learning framework

### 3.1 Motivation: Rank Matters in PEFT Methods

LoRA (Hu et al., 2022), Adapter (Li & Liang, 2021), and Prefix-Tuning (Houlsby et al., 2019) are the representative PEFT methods in the community. These PEFT methods share similar insights and can be unified (He et al., 2021). More specifically,

- The core idea of LoRA is proposing to model the incremental update of the pre-trained weights $W_{\text{pre-trained}} \in \mathbb{R}^{d_1 \times d_2}$ through low-rank approximations $\mathbf{z} = \mathbf{x}\,(W_{\text{pre-trained}} + \boldsymbol{B}\boldsymbol{A})$, where $\boldsymbol{B}\boldsymbol{A}$ is trainable and in low rank, $\mathbf{x} \in \mathbb{R}^{d_1}$, $\boldsymbol{B} \in \mathbb{R}^{d_1 \times r}$, $\boldsymbol{A} \in \mathbb{R}^{r \times d_2}$, and $r \ll \{d_1, d_2\}$.
- In Adapter or Prefix-Tuning, a module is inserted into a pre-trained model with structure $\mathbf{z} = \mathbf{x}W_{\text{pre-trained}} + f(\mathbf{x}\boldsymbol{B})\boldsymbol{A}$, where $\boldsymbol{B} \in \mathbb{R}^{d_1 \times r}$ and $\boldsymbol{A} \in \mathbb{R}^{r \times d_2}$ are trainable low-rank parameters. $\mathbf{x} \in \mathbb{R}^{d_1}$ is the input data, $f_{\text{non-linear}}(\cdot)$ is the non-linear function, and $r \ll \{d_1, d_2\}$.

While the compact dimensions of $\boldsymbol{B}$ and $\boldsymbol{A}$ offer notable parameter efficiency benefits, they also impose significant constraints on model capacity due to the inherent low-rank nature of their product, particularly when inner dimension $r$ is low. This restriction inevitably leads to suboptimal model performance, as evidenced by the findings presented in Figure 4 of He et al. (2021), Figure 2 of Zhang et al. (2022a), and our Figure 3(b).

To unlock superior performance, a crucial imperative emerges: elevating the rank of $\boldsymbol{B}\boldsymbol{A}$.

### 3.2 Increasing Model Capacity by Increasing the Rank

Hereby, we propose the following theorem which indicates an efficient potential to increase rank.

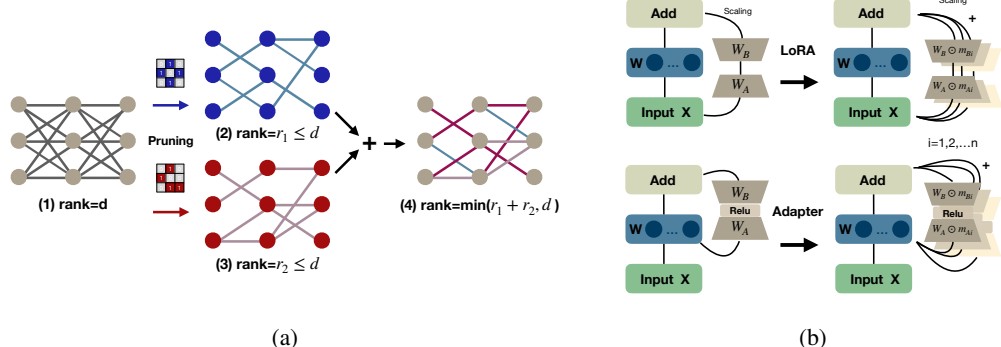

(a)             (b)

Figure 2: **The framework of CAPABOOST. (a): Diagram of CAPABOOST learning with d=2.** After applying blue and red pruning masks to the original $\mathbf{w}$ in (1), we obtain $\mathbf{w} \odot \mathbf{m}_{\text{blue}}$ in (2) and $\mathbf{w} \odot \mathbf{m}_{\text{red}}$ in (3), both sharing the same dense weight as (1). Since (2) and (3) have common pruned weights, we can exclude common pruned weights from the original $\mathbf{w}$ and store the sparse weights in (4), benefitting from fewer parameter numbers. During the training, we can retrieve weights from (4) and apply respective masks $\{\mathbf{m}_i\}$ to obtain weight in (2) and (3). **(b): Diagram of CAPABOOST example in LoRA and Adapter.**

**Theorem 3.1.** *Assume two matrices $\mathbf{X}$ and $\mathbf{Y}$ are randomly generated by $\mathbf{X} = \mathbf{X}^{col}\mathbf{X}^{row}$ and $\mathbf{Y} = \mathbf{Y}^{col}\mathbf{Y}^{row}$ respectively. $\mathbf{X}^{col} := [\mathbf{x}_1^{col}, \mathbf{x}_2^{col}, \dots, \mathbf{x}_r^{col}] \in \mathbb{R}^{d \times r}$, where column vector basis $\{\mathbf{x}_1^{col}, \mathbf{x}_2^{col}, \dots, \mathbf{x}_r^{col}\}$ are sampled from $\mathcal{N}(0, \mathbf{I}_d)$. Similarly, $\mathbf{X}^{row} = [\mathbf{x}_1^{row}, \mathbf{x}_2^{row}, \dots, \mathbf{x}_r^{row}]^\top \in \mathbb{R}^{r \times d}$ by sampling row vector basis $\{\mathbf{x}_1^{row}, \mathbf{x}_2^{row}, \dots, \mathbf{x}_r^{row}\}$ from $\mathcal{N}(0, \mathbf{I}_d)$. $\mathbf{I}_d \in \mathbb{R}^{d \times d}$ denotes an identity matrix. For matrices $\mathbf{X} \in \mathbb{R}^{d \times d}$ and $\mathbf{Y} \in \mathbb{R}^{d \times d}$, we have*

$$rank(\mathbf{X}+\mathbf{Y}) = rank(\mathbf{X})+rank(\mathbf{Y}) \text{ with probability equal to 1 almost surely when } 2r < d\,. \quad (1)$$

Proof of this theorem is provided in Appendix A.

**Remark 3.2.** *The matrix $\mathbf{X}$ in Theorem 3.1 is generated by the multiplication of two low-rank matrices, which share an identical form with trainable weight $\mathbf{w} = \mathbf{BA}$ in the PEFT methods. The vectors in $\mathbf{X}^{col}$ and $\mathbf{X}^{row}$ are sampled from Gaussian distribution, while $\mathbf{A}$ is also initialized in Gaussian. Hence, the high similarity between $\mathbf{w}$ and $\mathbf{X}$ suggests that this theorem might be applicable and effective in PEFT methods as well.*

## 3.3    CAPABOOST: INCREASING MODEL CAPACITY FOR FREE!

In addition to increasing the rank of low-rank-based PEFT methods to enhance model capacity, we explore a complementary approach in this section based on the insights from Theorem 3.1.

**A naive solution.** Theorem 3.1 intuitively motivates us to add more parallel trainable parameters. In detail, the capacity of a general linear layer $\mathbf{w}$ with $\mathbf{z} = \mathbf{w}\mathbf{x} + \mathbf{b}$ can be improved by:

$$\mathbf{z} = \sum_{i=1}^{d} \mathbf{w}_i\mathbf{x} + \mathbf{b} = \left(\tilde{\mathbf{w}} := \sum_{i=1}^{d} \mathbf{w}_i\right)\mathbf{x} + \mathbf{b}\,, \quad (2)$$

where $rank(\tilde{\mathbf{w}}) \cong d \cdot rank(\mathbf{w})$, once $\{\mathbf{w}_i\}$ can satisfy the distribution assumption in Theorem 3.1. However, this approach becomes impractical due to the $d-1$ times increase in the number of additional parameters, rendering PEFT methods inefficient. As a remedy, we introduce the CAPABOOST, a cost-free solution, detailed below.

**CAPABOOST framework.** We leverage the idea of weight-tying to alleviate the parameter-inefficient overhead caused by the naive solution discussed above. As intuitively visualized in Figure 2(a) and equations below, CAPABOOST constructs $\{\mathbf{w}_i\}$ for free using diverse pruning masks $\{\mathbf{m}_i\}$ with original weights $\mathbf{w}$:

$$\mathbf{z} = \sum_{i=1}^{d} (\mathbf{w} \odot \mathbf{m}_i)\mathbf{x} + \mathbf{b} = \left(\sum_{i=1}^{d} \mathbf{w} \odot \mathbf{m}_i\right)\mathbf{x} + \mathbf{b}\,, \quad (3)$$

where $d$ is the number of parallel tied models, and $\mathbf{m}_i$ is the corresponding pruning mask. Unless specified otherwise, we utilize a sparsity of $0.5$ in each $\mathbf{m}_i$ by default. These boolean masks $\{\mathbf{m}_i\}$ are distinct, static, and non-trainable across the training, and should be determined before the training phase. The storage overhead of these boolean masks can be avoided by using *a random generator* to generate deterministic masks on the fly with several scalar seeds per forward and backward pass.

**Benefits of CAPABOOST.** With pruning masks, $\mathbf{w} \odot \mathbf{m}_i$ and $\mathbf{w} \odot \mathbf{m}_j$ become different matrices and could potentially increase model rank following Theorem 3.1. The additional study shown in Appendix I suggests that the superiority of CAPABOOST cannot solely be attributed to the implicit regularization induced by the random masking, which emphasizes the advantage of boosting the model capacity. The CAPABOOST framework not only results in an increased model capacity but also allows a further reduction in the number of parameters and even FLOPs. The original parameter $\mathbf{w}$ will be pruned by a ratio of $1 - s^d$ with independently generated masks, where $s$ represents the sparsity ratio of masks. The pruning mask storage can also be avoided by the random generator mentioned above. An illustration is given in Figure 2(a).

Benefiting from the sparse matrix, the inference phase of CAPABOOST enjoys both reduced parameter numbers and FLOPs. Though the total FLOPS of CAPABOOST during the training is a factor of $s \times d$ to that of a dense layer, it could remain unchanged or lower than the dense one by appropriately selecting values for $s$ and $d$. Moreover, the computation involving sparse matrices can be accelerated using NVIDIA's sparse tensor core technology for both training and inference (Choquette et al., 2021; Zhang et al., 2022c). In Appendix H, we demonstrate that CAPABOOST can enjoy the hardware acceleration while preserving the performance gains from the increased model capacity.

Therefore, with proper selection of $s$ and $d$, we can implement CAPABOOST nearly for free as it can achieve fewer parameter numbers with similar FLOPs compared to original PEFT methods.

**Adapting CAPABOOST to PEFT methods.** In Figure 2(b), we illustrate how to utilize CAPABOOST as a plugin for various existing algorithms to further enhance the model capacity.

• CAPABOOST-LoRA (adapt $\mathbf{z} = \mathbf{x}(\boldsymbol{W}_{\text{pre-trained}} + \boldsymbol{B}\boldsymbol{A})$):

$$\mathbf{z} = \mathbf{x}\boldsymbol{W}_{\text{pre-trained}} + \mathbf{x}\left( \sum_{i=1}^{d} (\boldsymbol{B} \odot \mathbf{m}_{b_i})(\boldsymbol{A} \odot \mathbf{m}_{a_i}) \right). \tag{4}$$

• CAPABOOST-Adapter / CAPABOOST-Prefix-Tuning (adapt $\mathbf{z} = \mathbf{x}\boldsymbol{W}_{\text{pre-trained}} + f_{\text{non-linear}}(\mathbf{x}\boldsymbol{B})\boldsymbol{A}$):

$$\mathbf{z} = \mathbf{x}\boldsymbol{W}_{\text{pre-trained}} + \sum_{i=1}^{d} \left( f_{\text{non-linear}}\big(\mathbf{x}(\boldsymbol{B} \odot \mathbf{m}_{b_i})\big)\big(\boldsymbol{A} \odot \mathbf{m}_{a_i}\big) \right). \tag{5}$$

**Remark 3.3** (The consistency between intuition and practice). *To verify the idea of rank expansion, we measure the rank in LoRA and* CAPABOOST-*LoRA separately, denoted as* $\text{rank}(\boldsymbol{B}\boldsymbol{A})$ *and* $\text{rank}\left( \sum_{i=1}^{d} (\boldsymbol{B} \odot \mathbf{m}_{b_i})(\boldsymbol{A} \odot \mathbf{m}_{a_i}) \right)$. *Using RoBERTa-base model as the base model following settings in Zeng et al. (2023), we train LoRA and* CAPABOOST-*LoRA on the CoLA (i.e., GLUE subtask dataset (Wang et al., 2018)) and compare their ranks at the final epoch. The results in Table 1 consistently show that the rank of* CAPABOOST-*LoRA is always $d$ times of that in LoRA, regardless of any changes in the inner rank $r$. This observation confirms our theoretical intuition.*

## 4 EXPERIMENTAL SETUP

We briefly summarize the experimental setup in this section. More details can be found in Appendix B.

**Models and Datasets.** We conduct a broad range of experiments on various tasks, including (a) Natural Language Understanding (NLU), (b) Question Answering (QA), and (c) Computer Vision (CV). In NLU, we evaluate the fine-tuning performance of RoBERTa-base (Liu et al., 2019) and DeBERTaV3-base (He et al., 2022a) on GLUE (Wang et al., 2018) benchmark using a diverse array of PEFT algorithms. In QA, we evaluate the performance of the proposed PEFT method on SQuADv1.1 (Rajpurkar et al., 2016) and SQuADv2.0 (Rajpurkar et al., 2018). For the CV task, we use ViT-B/16 (Dosovitskiy et al., 2020) as our pre-trained model and evaluate the efficacy of our method on the VTAB-1k (Zhai et al., 2019) benchmark.

Table 1: **CAPABOOST-LoRA always has a higher rank than that of LoRA**. The rank is calculated in the RoBERTa-base model after training on the CoLA dataset using three different random seeds. $d$ represents the number of parallel tied modules, and $r$ represents the inner dimension of matrices $\boldsymbol{B}$ and $\boldsymbol{A}$. *#Param* is a factor to that of LoRA ($r = 8$).

| Rank of method | $r = 8$ | $r = 16$ | $r = 32$ | $r = 64$ |
|---|---|---|---|---|
| LoRA | 8 | 16 | 32 | 64 |
| #Param | 1x | 2x | 4x | 8x |
| CAPABOOST-LoRA ($d = 2$) | 16 | 32 | 64 | 128 |
| #Param | 0.75x | 1.5x | 3x | 6x |
| CAPABOOST-LoRA ($d = 4$) | 32 | 64 | 128 | 256 |
| #Param | 0.94x | 1.9x | 3.75x | 7.5x |

**Backbones.** For NLU tasks, similar to ProPETL (Zeng et al., 2023) and AdaLoRA (Zhang et al., 2022a), we implement CAPABOOST-LoRA and CAPABOOST-PAdapter for fine-tuning RoBERTa-base (Liu et al., 2019) and DeBERTaV3-base (He et al., 2022a) on GLUE. During fine-tuning, only parameters in incremental modules and the text classification head are trainable and the other model parameters remain frozen. For QA tasks, we use the publicly available fine-tuning example scripts

from *Adapter-Transformers* (Pfeiffer et al., 2020) library and test on DeBERTaV3-base (He et al., 2022a) model. For CV tasks, the ViT-B/16 (Dosovitskiy et al., 2020) model is used to evaluate the efficacy of our methods for experiments based on the VTAB-1k benchmark.

**Baselines.** We validate the effectiveness of our CAPABOOST framework by comparing it with the following baselines:

- *Fully fine-tuning* is the common practice for adaptation on downstream tasks. All model parameters are trainable during fine-tuning.
- *BitFit* (Zaken et al., 2022) only fine-tunes the bias vector in each layer of the model.
- *Prefix-Tuning* (Li & Liang, 2021) concatenates two sets of trainable prefix vectors to the keys and values in the attention layer, respectively. Then multi-head attention is computed based on the new prefixed keys and values.
- *Adapter tuning* inserts two adapter modules, which are connected consecutively by a nonlinear activation function, between transformer blocks. We consider *HAdapter* (Houlsby et al., 2019) and its popular variant *PAdapter* (Pfeiffer et al., 2021) as our baselines. The comparison of budget configuration between *HAdapter* and *PAdapter* can be found in Appendix D.
- *LoRA* (Hu et al., 2022) parameterizes incremental updates by two low-rank matrices. The number of trainable parameters is solely dependent on the preset rank value $r$ and the depth of the model $n$.
- *AdaLoRA* (Zhang et al., 2022a) emphasizes the varying importance of different weights and innovatively proposes to dynamically allocate the parameter budget among different layers during fine-tuning. It is worth noting that AdaLoRA inserts incremental modules into all weight matrices and achieves better performance compared to only fine-tuning queries and values in LoRA[1].
- *ProPETL* (Zeng et al., 2023) shares a single prototype PEFT module across all layers and tasks, and learns diverse binary masks that represent layer-specific and task-specific updates for adaptation. Since PEMN (Bai et al., 2022) shares the similar idea with ProPETL and the latter is the current state-of-the-art, we only use ProPETL as our baseline in this work.

**Setups.** To broadly compare with other approaches, we replicate the setups used by prior work and reuse reported optimal performance from the original papers as much as possible. We develop CAPA-BOOST-LoRA and CAPABOOST-PAdapter in the codebase provided by ProPETL (Zeng et al., 2023). In Table 2, we follow the evaluation setup as ProPETL (Zeng et al., 2023) on the RoBERTa-base model, which divides the original development set into two parts: evaluation set and test set. Model selection is performed on the evaluation set[2], and the final performance is reported on the test set. In Table 3, we adopt the evaluation strategy from AdaLoRA (Zhang et al., 2022a), which uses the original development set as the test set and skips model selection. For all baselines, we use the method-specific hyper-parameters reported in prior work and only tune the learning rate to get the best performance. Hyper-parameter tuning details defer to the Appendix. We use NVIDIA RTX-4090 for all experiments.

## 5 RESULTS

Our CAPABOOST is able to unleash the potential of PEFT methods across both NLP and CV domains, by increasing the model capacity without introducing additional parameters and computational cost.

### 5.1 NATURAL LANGUAGE UNDERSTANDING

**Fine-tuning RoBERTa-base on GLUE.** We summarize the performance of CAPABOOST-LoRA, CAPABOOST-PAdapter, and other baselines on the GLUE benchmark in Table 2, following the evaluation setup as *ProPETL* (Zeng et al., 2023) on RoBERTa-base model. *Both CAPABOOST-LoRA and CAPABOOST-PAdapter achieve better performance compared to their respective counterparts, while containing even less trainable parameters.* Specifically, (1) CAPABOOST-PAdapter significantly outperforms the original *PAdapter* (86.31 v.s. 84.30) in terms of the average score on the GLUE benchmark, while reducing the trainable parameters by 25.3% (0.71% v.s. 0.95%); (2) CAPABOOST-LoRA also improves the score of *LoRA* by 1.9 with the same extent of parameter reduction as CAPABOOST-PAdapter. These results reveal that CAPABOOST *randomly masking weights in parallel tied modules is capable of enhancing the fine-tuning performance on downstream tasks via increasing the actual rank of incremental weights.*

---

[1] For the sake of fair comparison with other methods, we append incremental weights to self-attention layers for AdaLoRA in Table 2. We reuse AdaLoRA results from the prior work (Zhang et al., 2022a) in Table 3, which adds incremental weights to all weight matrices.

[2] We modify the AdaLoRA codebase to facilitate evaluation under this setup, ensuring a fair comparison.

Table 2: **Performance of all PEFT methods based on RoBERTa-base on the GLUE tasks**. We use two parallel tied modules with 0.5 sparsity here in CAPABOOST-LoRA and CAPABOOST-PAdapter, in which the rank of each returned parameter matrices is doubled. The best results on each downstream task are shown in **bold**. Here *% FT Params* stands for the percentage of trainable parameters relative to that in fully fine-tuning. We report Matthew's Correlation for CoLA, average correlation for STS-B, and accuracy for the other tasks. Higher is better for all metrics. "*" indicates the results are reported in prior works. FLOPs are calculated as a factor relative to that of LoRA. Results are averaged over three trials. The details of budget configuration can be found in Appendix D.1.1.

| Methods | % FT Params | FLOPs | CoLA M corr. | SST-2 Acc. | MRPC Acc. | QQP Acc. | STS-B Corr. | MNLI Acc. | QNLI Acc. | RTE Acc. | All Avg. |
|---|---|---|---|---|---|---|---|---|---|---|---|
| Fine-tuning* | 100.00% | - | 60.94 | 94.04 | 87.25 | **91.34** | 90.83 | **87.57** | 92.53 | 73.14 | 84.71 |
| Prefix-Tuning* | 0.95% | - | 63.27 | 94.42 | 89.05 | 88.86 | 90.43 | 85.76 | 91.46 | 63.79 | 83.38 |
| ProPETL-Prefix-Tuning* | 1.03% | - | 61.81 | 94.00 | 87.42 | 88.85 | 90.48 | 85.73 | 91.05 | 63.79 | 82.89 |
| LoRA* | 0.95% | 1x | 62.24 | 93.81 | 86.76 | 88.79 | 90.61 | 86.59 | 91.85 | 67.63 | 83.54 |
| ProPETL-LoRA* | 1.04% | 1.02x | 62.16 | 93.62 | 88.73 | 87.59 | 90.88 | 85.30 | 91.75 | 72.66 | 84.09 |
| AdaLoRA | 1.07% | 1.44x | $63.86_{\pm0.7}$ | $94.11_{\pm0.1}$ | $88.24_{\pm1.1}$ | $90.03_{\pm0.1}$ | $91.36_{\pm0.2}$ | $87.08_{\pm0.1}$ | $92.33_{\pm0.1}$ | $76.49_{\pm2.0}$ | $85.44_{\pm0.2}$ |
| CAPABOOST-LoRA | **0.71%** | 1x | $63.74_{\pm0.6}$ | $94.19_{\pm0.1}$ | $88.24_{\pm0.0}$ | $90.88_{\pm0.1}$ | $91.52_{\pm0.2}$ | $87.40_{\pm0.3}$ | $92.60_{\pm0.2}$ | $75.06_{\pm2.4}$ | $85.45_{\pm0.3}$ |
| PAdapter* | 0.95% | 1x | 63.15 | 94.00 | 86.93 | 89.78 | 90.74 | 87.10 | 92.23 | 70.50 | 84.30 |
| ProPETL-PAdapter* | 1.04% | 1.03x | 65.43 | 94.15 | 88.24 | 89.40 | 91.14 | 86.53 | 92.58 | 76.50 | 85.50 |
| CAPABOOST-PAdapter | **0.71%** | 1x | $\mathbf{66.88}_{\pm0.5}$ | $\mathbf{94.50}_{\pm0.5}$ | $\mathbf{89.38}_{\pm1.2}$ | $91.13_{\pm0.0}$ | $91.09_{\pm0.1}$ | $87.30_{\pm0.1}$ | $92.59_{\pm0.2}$ | $\mathbf{77.94}_{\pm2.4}$ | $\mathbf{86.35}_{\pm0.4}$ |

Table 3: **Performance of all PEFT methods based on DeBERTaV3-base on the GLUE tasks**. We use two parallel tied modules with 0.5 sparsity here in CAPABOOST-LoRA and CAPABOOST-PAdapter, in which the rank of each returned parameter matrices is doubled. The best results on each downstream task are shown in **bold**. Here *% FT Params* stands for the percentage of trainable parameters relative to that in fully fine-tuning. We report Matthew's Correlation for CoLA, average correlation for STS-B, and accuracy for the other tasks. Higher is better for all metrics. "*" indicates the results are reported in prior works. FLOPs are calculated as a factor relative to that of LoRA. CAPABOOST-PAdapter only has a bottleneck dimension of 32 to simulate the constrained parameter budget situation, while the other PAdapter variants have a bottleneck dimension of 64. Results are averaged over three trials. More details of budget configuration can be found in Appendix D.2.1.

| Methods | % FT Params | FLOPs | CoLA M corr. | SST-2 Acc. | MRPC Acc. | QQP Acc. | STS-B Corr. | MNLI Acc. | QNLI Acc. | RTE Acc. | All Avg. |
|---|---|---|---|---|---|---|---|---|---|---|---|
| Fine-tuning* | 100.00% | - | 69.19 | 95.63 | 89.46 | **92.40** | 91.60 | 89.90 | 94.03 | 83.75 | 88.25 |
| BitFit* | **0.05%** | - | 66.96 | 94.84 | 87.75 | 88.41 | 91.35 | 89.37 | 92.24 | 78.70 | 86.20 |
| LoRA* | 0.72% | 1x | 69.82 | 94.95 | 89.95 | 91.99 | 91.60 | 90.65 | 93.87 | 85.20 | 88.50 |
| AdaLoRA* | 0.69% | 2.15x | 71.45 | 96.10 | 90.69 | 92.23 | 91.84 | 90.76 | **94.55** | **88.09** | 89.46 |
| ProPETL-LoRA | 0.78% | 1.02x | $69.12_{\pm0.8}$ | $95.72_{\pm0.2}$ | $90.44_{\pm0.3}$ | $91.10_{\pm0.1}$ | $91.52_{\pm0.1}$ | $90.29_{\pm0.1}$ | $94.28_{\pm0.1}$ | $85.20_{\pm0.8}$ | $88.46_{\pm0.2}$ |
| CAPABOOST-LoRA | 0.54% | 1x | $72.22_{\pm2.2}$ | $95.80_{\pm0.4}$ | $91.42_{\pm1.0}$ | $91.68_{\pm0.0}$ | $91.99_{\pm0.1}$ | $\mathbf{91.10}_{\pm0.0}$ | $93.83_{\pm0.2}$ | $86.64_{\pm0.3}$ | $89.33_{\pm0.3}$ |
| HAdapter* | 0.66% | 1.03x | 68.64 | 95.53 | 89.95 | 91.91 | 91.48 | 90.13 | 94.11 | 84.48 | 88.28 |
| PAdapter* | 0.64% | 0.99x | 68.77 | 95.61 | 89.46 | 92.04 | 91.54 | 90.33 | 94.29 | 85.20 | 88.41 |
| ProPETL-PAdapter | 0.69% | 1.02x | $70.47_{\pm0.8}$ | $95.99_{\pm0.1}$ | $90.44_{\pm0.3}$ | $91.74_{\pm0.0}$ | $91.48_{\pm0.0}$ | $90.46_{\pm0.1}$ | $94.32_{\pm0.0}$ | $85.92_{\pm0.3}$ | $88.90_{\pm0.1}$ |
| CAPABOOST-PAdapter | 0.24% | 0.5x | $\mathbf{73.13}_{\pm1.0}$ | $\mathbf{96.22}_{\pm0.2}$ | $\mathbf{92.16}_{\pm1.2}$ | $92.26_{\pm0.0}$ | $\mathbf{92.37}_{\pm0.2}$ | $90.12_{\pm0.1}$ | $94.12_{\pm0.1}$ | $87.05_{\pm1.8}$ | $\mathbf{89.68}_{\pm0.4}$ |

We also find the performance of CAPABOOST is competitive compared to state-of-the-art contenders: (i) CAPABOOST-PAdapter/CAPABOOST-LoRA surpasses over *ProPETL-PAdapter/ProPETL-LoRA* in 7 out of 8 downstream tasks; (ii) CAPABOOST-LoRA can perform just as well as AdaLoRA while using significantly less trainable parameters.

**Fine-tuning DeBERTaV3-base on GLUE.** We repeat experiments based on the DeBERTaV3-base model and report results on the GLUE development set in Table 3. *We observe that* CAPABOOST *achieves better or on-par overall performance on the GLUE benchmark compared with existing approaches.* We examine the capability of CAPABOOST in situations with constrained parameter budgets by cutting the bottleneck dimensions of CAPABOOST-PAdapter by half, leading to only 0.24% trainable parameters relative to that in the fully fine-tuning. The superior performance of CAPABOOST-PAdapter, despite having only half the bottleneck dimension compared to other baselines, highlights its ability to effectively leverage shared parameters in parallel tied modules and learn more powerful features for adaptation.

## 5.2 QUESTION ANSWERING

To verify CAPABOOST can also work in more difficult tasks, we evaluate our method on two QA datasets: SQuADv1.1 and SQuADv2.0, under four different budget settings: 0.08%, 0.16%, 0.32%, and 0.65%. The results are presented in Table 4. We find that

Table 4: **CAPABOOST-PAdapter results with DeBERTaV3-base on SQuAD v1.1 and SQuADv2.0.** We use two parallel tied modules with 0.5 sparsity here in CAPABOOST-PAdapter. Here *% FT Params* is the number of trainable parameters relative to that in full fine-tuning. We report EM/F1 for both tasks. The best results in each setting are shown in **bold**. Higher is better for all metrics. CAPABOOST-PAdapter achieves the best overall performance compared to other baselines, $1.0\%$ / $0.7\%$ higher than LoRA, the second best-performing method on SQuAD datasets, while containing less trainable parameters.

| | SQuADv1.1 | | | | SQuADv2.0 | | | | Avg |
|---|---|---|---|---|---|---|---|---|---|
| Full FT | 86.0 / 92.7 | | | | 85.4 / 88.4 | | | | 85.7 / 90.6 |
| % FT Params | 0.08% | 0.16% | 0.32% | 0.65% | 0.08% | 0.16% | 0.32% | 0.65% | - |
| HAdapter | 84.4/91.5 | 85.3/92.1 | 86.1/92.7 | 86.7/92.9 | 83.4/86.6 | 84.3/87.3 | 84.9/87.9 | 85.4/88.3 | 85.1/89.9 |
| PAdapter | 84.4/91.7 | 85.9/92.5 | 86.2/92.8 | 86.6/93.0 | 84.2/87.2 | 84.5/87.6 | 84.9/87.8 | 84.5/87.5 | 85.2/90.0 |
| LoRA | 86.4/92.8 | 86.6/92.9 | 86.7/93.1 | 86.7/93.1 | 84.7/87.5 | 83.6/86.7 | 84.5/87.4 | 85.0/88.0 | 85.5/90.2 |
| % FT Params | 0.06% | 0.12% | 0.24% | 0.49% | 0.06% | 0.12% | 0.24% | 0.49% | - |
| CAPABOOST-PAdapter | **87.6/93.5** | **87.3/93.4** | **87.4/93.4** | **87.7/93.5** | 85.1/87.9 | **85.5/88.5** | **85.8/88.6** | **85.8/88.6** | **86.5/90.9** |

Table 5: **CAPABOOST-LoRA for parameter-efficient transfer learning on VTAB-1k**. CAPABOOST-LoRA achieves better overall performance, $1\%$ *higher than LoRA*, while the parameter numbers remain the same. We follow the experimental settings in Zhang et al. (2022b). The base model used is ViT-B/16 pre-trained on ImageNet-22K, and LoRA/CAPABOOST-LoRA contain 0.29M/0.22M trainable parameters. Both LoRA and CA-PABOOST-LoRA have inner dimension $r = 8$ and are trained for 100 epochs. The result of VPT-Deep and NOAH are collected from Jia et al. (2022) and Zhang et al. (2022b) respectively. Results are averaged over three trials.

| Model Name | Natural | | | | | Specialized | | Structured | | | | | Average |
|---|---|---|---|---|---|---|---|---|---|---|---|---|---|
| | CIFAR-100 | DTD | Flower102 | Pets | SVHN | Eurosat | Resisc45 | Clevr-Count | Clevr-Dist | DMLab | dSpr-Ori | sNORB-Azim | |
| VPT-Deep | **78.80** | 65.80 | 98.00 | 88.30 | 78.10 | **96.10** | 83.40 | 68.50 | 60.00 | 46.50 | 47.90 | **32.90** | 70.36 |
| NOAH | 69.60 | 70.20 | **99.10** | 90.40 | 86.10 | 95.40 | 83.90 | **82.80** | 68.90 | 49.90 | **48.30** | 32.80 | 73.11 |
| LoRA | $67.10_{\pm 0.3}$ | $69.89_{\pm 0.5}$ | $98.95_{\pm 0.1}$ | $90.58_{\pm 0.3}$ | $84.20_{\pm 0.7}$ | $95.34_{\pm 0.2}$ | $85.93_{\pm 0.1}$ | $82.68_{\pm 0.3}$ | $67.92_{\pm 0.5}$ | $49.68_{\pm 0.3}$ | $45.54_{\pm 0.2}$ | $30.80_{\pm 0.4}$ | 72.38 |
| ProPETL-LoRA | $69.50_{\pm 0.4}$ | $70.48_{\pm 0.5}$ | $98.91_{\pm 0.0}$ | $90.84_{\pm 0.2}$ | $85.64_{\pm 0.9}$ | $95.41_{\pm 0.3}$ | $85.35_{\pm 0.1}$ | $81.99_{\pm 0.2}$ | $67.26_{\pm 0.4}$ | $49.79_{\pm 0.2}$ | $46.97_{\pm 0.5}$ | $28.74_{\pm 0.3}$ | 72.57 |
| CAPABOOST-LoRA (d=2) | $69.28_{\pm 0.5}$ | $71.45_{\pm 0.3}$ | $99.06_{\pm 0.0}$ | **$91.21_{\pm 0.3}$** | $85.01_{\pm 1.1}$ | $95.55_{\pm 0.2}$ | **$86.48_{\pm 0.1}$** | **$82.80_{\pm 0.2}$** | $69.05_{\pm 0.4}$ | $50.58_{\pm 0.8}$ | $46.04_{\pm 0.1}$ | $31.94_{\pm 0.2}$ | **73.20** |
| CAPABOOST-LoRA (d=4) | $69.01_{\pm 0.1}$ | **$72.18_{\pm 0.6}$** | $99.06_{\pm 0.0}$ | $91.20_{\pm 0.2}$ | **$87.07_{\pm 0.7}$** | $95.76_{\pm 0.1}$ | $85.96_{\pm 0.2}$ | $82.77_{\pm 0.3}$ | **$69.11_{\pm 0.3}$** | **$50.90_{\pm 0.9}$** | $45.84_{\pm 0.3}$ | $32.09_{\pm 0.3}$ | **73.41** |

- CAPABOOST-*PAdapter outperforms other baselines, including fully fine-tuning with all model parameters, in all budget settings except for a* $0.08\%$ *parameter budget on SQuADv2*. Moreover, it achieves this superiority while being more parameter-efficient, requiring $25\%$ fewer parameters. The average score improvement compared to the second best-performing method, LoRA, is 1.0/0.7.
- *HAdapter* and *PAdapter* degrade dramatically when reducing the number of trainable parameters on SQuADv1.1, while our method shows consistent performance under different budget levels.
- In the most constrained budget setting on SQuADv2, a more challenging dataset, CAPABOOST-PAdapter falls short of surpassing fully fine-tuning. We believe that in resource-constrained scenarios for demanding tasks, existing PEFT methods require more efficient parameter allocation across different layers. We leave the adaptive allocation of CAPABOOST over layers for future work.

## 5.3 VISUAL TASK ADAPTATION BENCHMARK

We further evaluate the efficacy of CAPABOOST on the benchmark with a different modality, VTAB-1k (Zhai et al., 2019), which includes several visual tasks from three categories: natural, specialized, and structured. Experimental results summarized in Table 5 show that *our method(d=2) brings consistent improvements over LoRA on all downstream tasks, leading to an average improvement of around* $1\%$ *on VTAB-1k, while maintaining the same or less storage and computation efficiency.* More parallel tied modules achieve larger performance gain from 73.20 to 73.41 as evidenced in Table 5, which shows vision tasks enjoy more benefits from higher rank updates.

## 6 DISCUSSION

In this section, we present ablation study results based on the CoLA dataset to validate the effectiveness of our method. Additional ablation studies on the SST-2 dataset can be found in Appendix G.

**Different mask sparsity.** Figure 3(a) illustrates the results of fine-tuning the RoBERTa-base model with different pruning mask sparsity on CoLA. Notably, we observe that CAPABOOST-LoRA reaches peak performance when the density of random masks equals 0.6. This observation highlights the

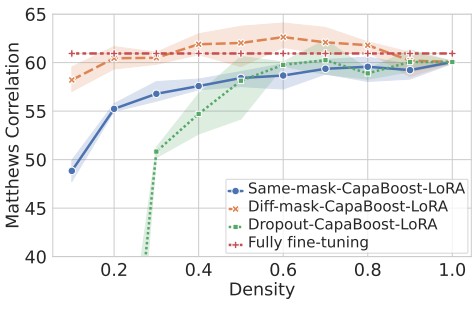 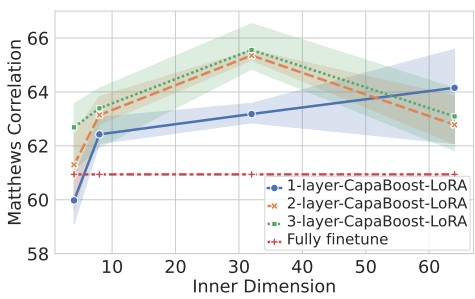

(a) Mask Density  (b) Inner Dimension $r$

Figure 3: **Ablation study for components of CAPABOOST-LoRA on RoBERTa-base model. (a)** Average performance of CAPABOOST-LoRA with different pruning masks, same pruning mask, and only Dropout without pruning over different sparsity on CoLA dataset. We use two parallel tied modules with a preset LoRA inner dimension of 8. **(b)** Average performance of CAPABOOST-LoRA under different rank values and number of parallel tied modules when density= 0.5 on CoLA dataset. Results are averaged over three trials.

crucial role of random masks in CAPABOOST, as they enforce non-linearity among parallel layers and effectively increase the actual rank of weight matrices, thereby enhancing the performance.

However, we observe a continuous degradation in the performance of CAPABOOST-LoRA when we vary the sparsity level in each parallel tied module. We attribute this trend to the diminishing effect of model capacity enhancement when the sparsity level becomes either too high or too low. Specifically, when the density is set to 1.0, CAPABOOST-LoRA is equivalent to the original LoRA multiplied by a coefficient. Conversely, at a density of 0.1, the rank of each parallel weight matrix is insufficient to effectively enhance model capacity.

**On the influence of mask types.** Dropout (Hinton et al., 2012) addresses overfitting by randomly setting parameters or activations to zero (Wan et al., 2013; Kingma et al., 2015; Gal & Ghahramani, 2016), similar to our masking to some extent. However, our method aims to boost the capacity of dense weights through learning with deterministic binary masks, rather than relying on regularization.

Figure 3(a) validates the efficacy of our design choice, *Diff-mask*, by comparing it with two alternatives: *Same-mask* and *Dropout*. Our findings reveal that *Diff-mask* consistently outperforms both alternatives across different sparsity levels, suggesting that different random but deterministic masks in CAPABOOST play vital roles in the performance gain. In the case of employing new random weight masking in each iteration like *Dropout*, it does result in poor performance when the density is low.

**Tradeoff between parameter budget and the number of parallel tied modules.** As discussed in Section 3.1, the rank of incremental weight matrix bottlenecks the model capacity, especially in the case with a limited parameter budget. We illustrate this point by conducting an ablation study over the inner dimension $r$ of LoRA modules as shown in Figure 3(b). We find that

- The performance of CAPABOOST-LoRA with one single layer on CoLA monotonically increases with the inner dimension $r$. It is aligned with our intuition in Section 3.1 that the performance is limited by the rank of incremental weight matrices when the model capacity is insufficient.
- CAPABOOST-LoRA with 2 or 3 parallel layers significantly outperforms the case with only one parallel layer. Specifically, the fine-tuning performance of 2-layer-CAPABOOST-LoRA with $r = 32$ surpasses that of 1-layer-CAPABOOST-LoRA with $r = 64$, where the former one has 25% fewer trainable parameters than that of 1-layer-CAPABOOST-LoRA.

These results demonstrate that the fine-tuning performance is bottlenecked by the model capacity and CAPABOOST can alleviate this problem by increasing the model capacity for free. However, we also notice that the performance of CAPABOOST-LoRA starts to degrade as we further increase the rank of incremental weights, which might be attributed to potential optimization difficulties e.g. overfitting.

**Limitations and future work.** Although CAPABOOST has demonstrated its prowess in enhancing model capacity, it becomes susceptible to overfitting when the internal dimension is set to a large value. Addressing this issue through adaptive weight allocation across layers remains a promising avenue for future research. Furthermore, CAPABOOST's remarkable parameter efficiency positions it as an ideal solution for tackling demanding applications, like fine-tuning Large Language Models. We also defer exploration of this potential to future work.

## ACKNOWLEDGEMENT

We thank anonymous reviewers for their constructive and helpful reviews. We also thank Mr. Shuailong Zhu for his great help in mathematical proof. This work was supported in part by the National Science and Technology Major Project (No. 2022ZD0115101), the Research Center for Industries of the Future (RCIF) at Westlake University, and the Westlake Education Foundation. Tao Lin and Soumajit Majumder were supported by an Industrial Research Grant from the Huawei Cloud Intelligent Cloud Technologies Initiative when initializing this project.

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

## CONTENTS OF APPENDIX

## A  PROOF OF THEOREM 3.1

We first rewrite the theorem:

**Theorem A.1.** *Assume two matrices $X$ and $Y$ are randomly generated by $X = X^{col} X^{row}$ and $Y = Y^{col} Y^{row}$ respectively. $X^{col} := [\mathbf{x}_1^{col}, \mathbf{x}_2^{col}, \dots, \mathbf{x}_r^{col}] \in \mathbb{R}^{d \times r}$, where column vector basis $\{\mathbf{x}_1^{col}, \mathbf{x}_2^{col}, \dots, \mathbf{x}_r^{col}\}$ are sampled from $\mathcal{N}(0, \mathbf{I}_d)$. Similarly, $X^{row} = [\mathbf{x}_1^{row}, \mathbf{x}_2^{row}, \dots, \mathbf{x}_r^{row}]^\top \in \mathbb{R}^{r \times d}$ by sampling row vector basis $\{\mathbf{x}_1^{row}, \mathbf{x}_2^{row}, \dots, \mathbf{x}_r^{row}\}$ from $\mathcal{N}(0, \mathbf{I}_d)$. $\mathbf{I}_d \in \mathbb{R}^{d \times d}$ denotes an identity matrix. For matrices $X \in \mathbb{R}^{d \times d}$ and $Y \in \mathbb{R}^{d \times d}$, we have*

$$rank(X+Y) = rank(X) + rank(Y) \text{ with probability equal to 1 almost surely when } 2r < d. \quad (6)$$

We rely on the following two lemmas to finish our proof.

**Lemma A.2** (Marsaglia (1964)). *Let $X$ and $Y$ be two matrices of the same size, $R_1$ and $R_2$ are their row spaces, $C_1$ and $C_2$ are their column spaces, $\cap$ is the linear space intersection. The rank of two matrices sum can be bound as:*

$$rank(X + Y) \geq rank(X) + rank(Y) - \dim(R_1 \cap R_2) - \dim(C_1 \cap C_2),$$

**Lemma A.3** ((Bogachev & Ruas, 2007)). *If $r < d$, then every $r$-dimensional subspace of $\mathbb{R}^n$ has $d$-dimensional Lebesgue measure zero.*

*Proof.* **Step (1).** Fix $X$, define column space of $X^{col}$ as

$$\text{ColumnSpace}(X^{col}) = \{\mathbf{x} \in \mathbb{R}^d \mid \mathbf{x} = \sum \lambda_i \mathbf{x}_i^{col}, \lambda_i \in \mathbb{R}\}. \quad (7)$$

For $\boldsymbol{X}$, let $\boldsymbol{X} = [\mathbf{x}_1, \mathbf{x}_2, \ldots, \mathbf{x}_r]$, then $\mathbf{x}_k = x_{k1}^{\text{row}}\mathbf{x}_1^{\text{col}} + x_{k2}^{\text{row}}\mathbf{x}_2^{\text{col}} + \ldots + x_{kr}^{\text{row}}\mathbf{x}_r^{\text{col}}$, which is a linear combination of $\{\mathbf{x}_1^{col}, \mathbf{x}_2^{col}, \ldots, \mathbf{x}_r^{col}\}$. Therefore, we have

$$\text{ColumnSpace}(\boldsymbol{X}) \subset \text{ColumnSpace}(\boldsymbol{X}^{\text{col}}) = \{x \in \mathbb{R}^d \mid \mathbf{x} = \sum \lambda_i \mathbf{x}_i^{\text{col}}, \lambda_i \in \mathbb{R}\}. \tag{8}$$

We want to show that,

$$\Pr\left(\text{ColumnSpace}(\boldsymbol{X}) \cap \text{ColumnSpace}(\boldsymbol{Y}) \neq \varnothing\right) = 0 \tag{9}$$

Since $\text{rank}(\boldsymbol{X}) \leq r$, $\text{ColumnSpace}(\boldsymbol{X})$ is a space built on the $r$ basis vectors and thus a proper subspace of dimension at most $r$.

We denote the probability measure with Gaussian distribution as $\mu$, which is continuous with respect to Lebesgue measure $dx$.

$$\Pr\left(\mathbf{y}_1^{\text{col}}\text{is linear dependent with}\{\mathbf{x}_1^{\text{col}}, ..., \mathbf{x}_r^{\text{col}}\}\right) = \mu\left(\text{ColumnSpace}(\boldsymbol{X})\right) = 0. \tag{10}$$

according to Lemma A.3.
Similarly, for vector $\mathbf{y}_j^{\text{col}}, \forall j > 1$, since $\{\mathbf{x}_1^{\text{col}}, \ldots, \mathbf{x}_r^{\text{col}}, \mathbf{y}_1^{\text{col}}, \ldots, \mathbf{y}_{j-1}^{\text{col}}\}$ has at most $r + j - 1 < d$ basis vectors, then

$$\begin{aligned}&\Pr\left(\mathbf{y}_j^{\text{col}}\text{is linear dependent with}\{\mathbf{x}_1^{\text{col}}, \ldots, \mathbf{x}_r^{\text{col}}, \mathbf{y}_1^{\text{col}}, \ldots, \mathbf{y}_{j-1}^{\text{col}}\}\right)\\&= \mu\left(\text{ColumnSpace}(\boldsymbol{X}) \cup \text{ColumnSpace}\left(\{\mathbf{y}_1^{\text{col}}, \ldots, \mathbf{y}_{j-1}^{\text{col}}\}\right)\right) = 0.\end{aligned} \tag{11}$$

Since all $\mathbf{y}_j^{\text{col}}$ is linearly dependent with previous vectors with probability 0, we can easily show Equation 9.

**Step (2).** Using the same argument, define the

$$\text{RowSpace}(\boldsymbol{X}) = \{\mathbf{x} \in \mathbb{R}^d \mid \mathbf{x} = \sum \lambda_i \mathbf{x}_i^{\text{row}}, \lambda_i \in \mathbb{R}\}, \tag{12}$$

it is easy to prove that,

$$\Pr\left(\text{RowSpace}(\boldsymbol{X}) \cap \text{RowSpace}(\boldsymbol{Y}) \neq \varnothing\right) = 0. \tag{13}$$

**Step (3).** With $\Pr\left(\text{ColumnSpace}(\boldsymbol{X}) \cap \text{ColumnSpace}(\boldsymbol{Y}) \neq \varnothing\right) = 0$ and $\Pr\left(\text{RowSpace}(\boldsymbol{X}) \cap \text{RowSpace}(\boldsymbol{Y}) \neq \varnothing\right) = 0$, we can get $\Pr\left(\dim(\boldsymbol{R}_1 \cap \boldsymbol{R}_2)\right) = 0$ and $\Pr\left(\dim(\boldsymbol{C}_1 \cap \boldsymbol{C}_2)\right) = 0$.
By Lemma A.2, $\Pr\left(\text{rank}(\boldsymbol{X} + \boldsymbol{Y}) \geq \text{rank}(\boldsymbol{X}) + \text{rank}(\boldsymbol{Y})\right) = 1$. Since $\text{rank}(\boldsymbol{X} + \boldsymbol{Y}) \leq \text{rank}(\boldsymbol{X}) + \text{rank}(\boldsymbol{Y})$(Marsaglia, 1964), $\Pr\left(\text{rank}(\boldsymbol{X} + \boldsymbol{Y}) = \text{rank}(\boldsymbol{X}) + \text{rank}(\boldsymbol{Y})\right) = 1$ □

**Remark A.4.** *The condition $2r < d$ exists because if $2r > d$ then Equation 11 doesn't hold since $r + j - 1 \geq d$ when $j = r$. Physically, since $\text{rank}(\boldsymbol{X} + \boldsymbol{Y}) \leq \dim(\boldsymbol{X} + \boldsymbol{Y}) = d$, it is impossible that $\text{rank}(\boldsymbol{X} + \boldsymbol{Y}) = 2r > d$*

# B  IMPLEMENTATION DETAILS

The implementation of our algorithm builds on the publicly available *PyTorch* (Paszke et al., 2019) and *Adapter-Transformers* (Pfeiffer et al., 2020).

# C  DATASETS

## C.1  GLUE BENCHMARK

The General Language Understanding Evaluation (GLUE) is a benchmark of nine sentence- or sentence-pair language understanding tasks (Wang et al., 2018), designed to evaluate and analyze the performance of language model with respect to a broad range of linguistic phenomena found in natural language tasks. Similar to two most recent work (Zeng et al., 2023; Zhang et al., 2022a), we conduct extensive experiments on 8 GLUE tasks, including CoLA, SST-2, MRPC, QQP, STS-B, MNLI, QNLI, and RTE. We present the dataset statistics of GLUE (Wang et al., 2018) in Table 6.

Table 6: Summary of the GLUE benchmark.

| **Corpus** | Task | #Train | #Dev | #Test | #Label | Metrics |
|---|---|---|---|---|---|---|
| | | Single-Sentence Classification (GLUE) | | | | |
| CoLA | Acceptability | 8.5k | 1k | 1k | 2 | Matthews corr |
| SST-2 | Sentiment | 67k | 872 | 1.8k | 2 | Accuracy |
| | | Pairwise Text Classification (GLUE) | | | | |
| MNLI | NLI | 393k | 20k | 20k | 3 | Accuracy |
| RTE | NLI | 2.5k | 276 | 3k | 2 | Accuracy |
| QQP | Paraphrase | 364k | 40k | 391k | 2 | Accuracy/F1 |
| MRPC | Paraphrase | 3.7k | 408 | 1.7k | 2 | Accuracy/F1 |
| QNLI | QA/NLI | 108k | 5.7k | 5.7k | 2 | Accuracy |
| | | Text Similarity (GLUE) | | | | |
| STS-B | Similarity | 7k | 1.5k | 1.4k | 1 | Pearson/Spearman corr |

Table 7: Statistics of the SQuAD dataset.

| | # Train | # Validation |
|---|---|---|
| SQuAD v1.1 | 87,599 | 10,570 |
| SQuAD v2.0 | 130,319 | 11,873 |

## C.2 SQuADv1.1 and SQuADv2.0

Stanford Question Answering Dataset (SQuAD) is a reading comprehension dataset, consisting of questions collected by crowdworkers on a set of Wikipedia articles, where the answer to every question is a segment of text, or span, from the corresponding reading passage, or the question might be unanswerable. The statistics of SQuADv1.1 and SQuADv2.0 are shown in Table 7.

## C.3 VTAB-1K

Visual Task Adaptation Benchmark (VTAB) proposed by Zhai et al. (2019), has good representations as those that adapt to diverse, unseen tasks with few examples. It includes CIFAT-100, DTD, Flower102, Pets, SVHN, Eurosat, Resisc45, Clevr-Count, Clevr-Dist, DMLab, dSpr-Ori and sNORB-Azim datasets and classify them into Natural, Specialized and Structured catagories.

# D    Natural Language Understanding

## D.1 RoBERTa-base

RoBERTa (Liu et al., 2019) uses an optimized pre-training recipe based on the one originally proposed in BERT (Devlin et al., 2018) and improves the performance on the same tasks with BERT without introducing many more tunable model parameters. It is a competitive choice of pre-trained model and broadly used by prior work for comprehensive evaluations (Hu et al., 2022; Zeng et al., 2023). We take the pre-trained RoBERTa-base model from the *HuggingFace Transformers* library (Wolf et al., 2020) and fine-tune it on the GLUE benchmark (Wang et al., 2018). We initialize the model to the pre-trained model at the beginning of every fine-tuning task.

### D.1.1 Budget Configuration

We present the budget configuration for experiments on RoBERTa-base in Table 8. $l/r$/bn control the parameter budget for prefix-tuning/LoRA/PAdapter-tuning, respectively. *% FT Params* denotes the percentage of fine-tunable parameters in the model. We use the same hyper-parameters for all tasks as prior work (Zeng et al., 2023) and show the difference in the percentage of trainable parameters relative to the fully fine-tuning.

### D.1.2 Training Details

We tune the learning rate from $\{1 \times 10^{-4}, 3 \times 10^{-4}, 5 \times 10^{-4}, 6 \times 10^{-4}, 1 \times 10^{-3}, 1.5 \times 10^{-3}, 2 \times 10^{-3}, 3 \times 10^{-3}, 5 \times 10^{-3}\}$ and pick the learning rate with the best performance for each dataset. The

***Prefix-tuning***

| PEFT Methods | Prefix | ProPETL | | |
|---|---|---|---|---|
| # Dimension ($l$) | 64 | 64 | - | - |
| % FT Params | 0.95% | 1.03% | - | - |

***PAdapter-tuning***

| PEFT Methods | PAdapter | ProPETL | CAPABOOST | |
|---|---|---|---|---|
| # Dimension (bn) | 64 | 64 | 64 | - |
| % FT Params | 0.95% | 1.04% | **0.71%** | - |

***LoRA***

| PEFT Methods | LoRA | ProPETL | CAPABOOST | AdaLoRA |
|---|---|---|---|---|
| # Dimension ($r$) | 32 | 32 | 32 | 32 |
| % FT Params | 0.95% | 1.04% | **0.71%** | 0.91% |

Table 8: Budget configuration for experiments on RoBERTa-base. $l$ denotes the length of prepended prefix vectors in Prefix-tuning. $r$ represents the preset rank value of LoRA modules. bn refers to the bottleneck dimension of Adapter modules.

details of hyper-parameters are shown in Table 9 and Table 10. For fair comparison, we only append incremental weights to self-attention layers in AdaLoRA as the other baselines do. We set the initial rank of each incremental matrix as 48 and the average target rank as 32.

Table 9: Hyper-parameter setup of CAPABOOST-LoRA for RoBERTa-base model on GLUE benchmark.

| Dataset | learning rate | batch size | # epochs | # tied layers | Density |
|---|---|---|---|---|---|
| **MNLI** | $5 \times 10^{-4}$ | 128 | 20 | 2 | 0.5 |
| **RTE** | $6 \times 10^{-4}$ | 128 | 40 | 2 | 0.5 |
| **QNLI** | $5 \times 10^{-4}$ | 128 | 10 | 2 | 0.5 |
| **MRPC** | $5 \times 10^{-4}$ | 128 | 40 | 2 | 0.5 |
| **QQP** | $5 \times 10^{-4}$ | 128 | 20 | 2 | 0.5 |
| **SST-2** | $1 \times 10^{-4}$ | 128 | 20 | 2 | 0.5 |
| **CoLA** | $5 \times 10^{-4}$ | 128 | 40 | 2 | 0.5 |
| **STS-B** | $5 \times 10^{-4}$ | 128 | 20 | 2 | 0.5 |

Table 10: Hyper-parameter setup of CAPABOOST-PAdapter for RoBERTa-base model on GLUE benchmark.

| Dataset | learning rate | batch size | # epochs | # tied layers | Density |
|---|---|---|---|---|---|
| **MNLI** | $1 \times 10^{-3}$ | 128 | 10 | 2 | 0.5 |
| **RTE** | $1 \times 10^{-3}$ | 128 | 40 | 2 | 0.5 |
| **QNLI** | $2 \times 10^{-3}$ | 128 | 10 | 2 | 0.5 |
| **MRPC** | $3 \times 10^{-4}$ | 128 | 30 | 2 | 0.5 |
| **QQP** | $1 \times 10^{-3}$ | 128 | 10 | 2 | 0.5 |
| **SST-2** | $1 \times 10^{-3}$ | 128 | 10 | 2 | 0.5 |
| **CoLA** | $5 \times 10^{-4}$ | 128 | 20 | 2 | 0.5 |
| **STS-B** | $1 \times 10^{-3}$ | 128 | 20 | 2 | 0.5 |

### D.2 DEBERTAV3-BASE

DeBERTa (He et al., 2020) proposes disentangled attention and enhanced mask decoder to enhance model performance and outperforms BERT (Devlin et al., 2018) and RoBERTa (Liu et al., 2019) on a majority of NLU tasks with 80GB training data. DeBERTaV3 further improves the training efficiency through using ELECTRA-Style (Clark et al., 2019) pre-training with Gradient Disentangled Embedding Sharing and achieves superior performance on downstream tasks. We take the pre-trained DeBERTaV3-base model from the *HuggingFace Transformers* library (Wolf et al., 2020) and fine-tune it on the GLUE benchmark (Wang et al., 2018). We initialize the model to the pre-trained model at the beginning of every fine-tuning task.

### D.2.1 BUDGET CONFIGURATION

We present the budget configuration for experiments on DeBERTaV3-base in Table 11. $r$/bn control the parameter budget for LoRA/PAdapter-tuning, respectively. % *FT Params* denotes the percentage of fine-tunable parameters in the model. We use the same experimental setup for all tasks as prior work (Zhang et al., 2022a) and show the difference in the percentage of trainable parameters relative to the fully fine-tuning. Note that AdaLoRA adds LoRA modules to all weight matrices of the model, while other LoRA-related variants only adds incremental modules to self-attention layers. Therefore, AdaLoRA has a smaller per-module rank value. We manually set a smaller bottleneck dimension for CAPABOOST-PAdapter here to examine its ability in the constrained budget setting.

| *Adapter-tuning* | | | | |
|---|---|---|---|---|
| PEFT Methods | PAdapter | ProPETL | CAPABOOST | HAdapter |
| # Dimension (bn) | 64 | 64 | 32 | 64 |
| % FT Params | 0.64% | 0.69% | **0.24%** | 0.66% |
| *LoRA* | | | | |
| PEFT Methods | LoRA | ProPETL | CAPABOOST | AdaLoRA |
| # Dimension ($r$) | 8 | 8 | 8 | 2 |
| % FT Params | 0.72% | 0.78% | **0.54%** | 0.69% |

Table 11: Budget configuration for experiments on DeBERTaV3-base. $r$ stands for the preset rank value of LoRA modules. bn refers to the bottleneck dimension of Adapter modules.

### D.2.2 TRAINING DETAILS

We tune the learning rate from $\{1 \times 10^{-4}, 3 \times 10^{-4}, 5 \times 10^{-4}, 8 \times 10^{-4}, 1 \times 10^{-3}, 1.5 \times 10^{-3}, 2 \times 10^{-3}, 3 \times 10^{-3}, 5 \times 10^{-3}\}$ and pick the learning rate with the best performance for each dataset. The details of hyper-parameters are shown in Table 12 and Table 13.

Table 12: Hyper-parameter setup of CAPABOOST-LoRA for DeBERTaV3-base model on GLUE benchmark.

| Dataset | learning rate | batch size | # epochs | # tied layers | Density |
|---|---|---|---|---|---|
| **MNLI** | $5 \times 10^{-4}$ | 128 | 20 | 2 | 0.5 |
| **RTE** | $2 \times 10^{-3}$ | 128 | 50 | 2 | 0.5 |
| **QNLI** | $1 \times 10^{-3}$ | 128 | 15 | 2 | 0.5 |
| **MRPC** | $3 \times 10^{-3}$ | 128 | 50 | 2 | 0.5 |
| **QQP** | $5 \times 10^{-4}$ | 128 | 20 | 2 | 0.5 |
| **SST-2** | $3 \times 10^{-4}$ | 128 | 25 | 2 | 0.5 |
| **CoLA** | $5 \times 10^{-4}$ | 128 | 15 | 2 | 0.5 |
| **STS-B** | $5 \times 10^{-4}$ | 128 | 25 | 2 | 0.5 |

Table 13: Hyper-parameter setup of CAPABOOST-PAdapter for DeBERTaV3-base model on GLUE benchmark.

| Dataset | learning rate | batch size | # epochs | # tied layers | Density |
|---|---|---|---|---|---|
| **MNLI** | $5 \times 10^{-4}$ | 128 | 20 | 2 | 0.5 |
| **RTE** | $1.5 \times 10^{-3}$ | 128 | 50 | 2 | 0.5 |
| **QNLI** | $5 \times 10^{-3}$ | 128 | 20 | 2 | 0.5 |
| **MRPC** | $5 \times 10^{-3}$ | 128 | 50 | 2 | 0.5 |
| **QQP** | $1 \times 10^{-3}$ | 128 | 30 | 2 | 0.5 |
| **SST-2** | $3 \times 10^{-4}$ | 128 | 10 | 2 | 0.5 |
| **CoLA** | $3 \times 10^{-4}$ | 128 | 10 | 2 | 0.5 |
| **STS-B** | $1 \times 10^{-3}$ | 128 | 20 | 2 | 0.5 |

# E    QUESTION ANSWERING

## E.1    BUDGET CONFIGURATION

We present the budget configuration for experiments on DeBERTaV3-base in Table 14.

Table 14: Detailed budget setup for question answering.

| # Params | Houlsby Adapter $d$ | Pfeiffer Adapter $d$ | LoRA $r$ | CAPABOOST-PAdapter $d$ |
|---|---|---|---|---|
| 0.65% | 32 | 64 | 8 | 64 |
| 0.32% | 16 | 32 | 4 | 32 |
| 0.16% | 8 | 16 | 2 | 16 |
| 0.08% | 4 | 8 | 1 | 8 |

## E.2    TRAINING DETAILS

We tune the learning rate from $\{1 \times 10^{-4}, 3 \times 10^{-4}, 5 \times 10^{-4}, 8 \times 10^{-4}, 1 \times 10^{-3}\}$ and pick the learning rate with the best performance for each dataset. The details of hyper-parameters are shown in Table 15.

Table 15: Hyper-parameter setup of CAPABOOST for DeBERTaV3-base model on question answering tasks.

| Dataset | learning rate | batch size | # epochs | # tied layers | Density |
|---|---|---|---|---|---|
| **SQuADv1.1** | $5 \times 10^{-4}$ | 64 | 15 | 2 | 0.5 |
| **SQuADv2.0** | $5 \times 10^{-4}$ | 64 | 15 | 2 | 0.5 |

# F    VISUAL TASK ADAPTATION

For the VTAB-1k(Zhai et al., 2019), we follows its default augmentation settings. All input images are resized to $224 \times 224$ and apply normalization with ImageNet means and standard deviation. The inner dimension of LoRA is set to 8.

# G    ADDITIONAL ABLATION STUDY

In this section, we implement additional ablation studies on SST-2-10k dataset, which contains 10k data examples uniformly sampled from the original SST-2 dataset, to further analyze the effectiveness of our proposed method. We use the same hyper-parameters as in section 6 for all experiments in this section. The results are shown in Figure 4.

# H    HARDWARE ACCELARATION UTILIZATION

The hardware acceleration for sparse matrix operation in CAPABOOST relies on NVIDIA's sparse tensor core technology. This requires sparse matrix(e.g. pruning mask in CAPABOOST) in a shape of N:M sparsity, for example, 2:4 sparsity. To verify our proposed CAPABOOST still enjoys performance benefits in N:M sparsity matrix, we present GLUE benchmark experimental results in Table 16.

The results indicate that CapaBoost performs similarly with both fully random masks and N:M sparsity random masks. Thus, we can directly replace fully random masks with N:M sparsity random masks and benefit from the hardware acceleration provided by the Nvidia sparse tensor core. In our experiment, the fine-tuned weight matrix with N:M sparsity displays the same rank as with fully random masks.

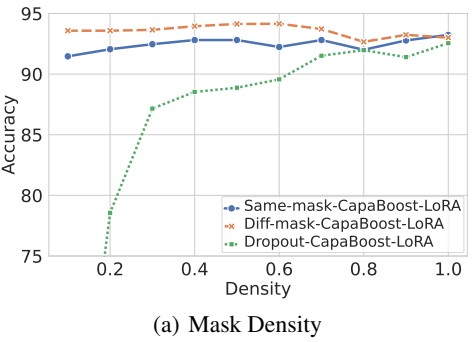 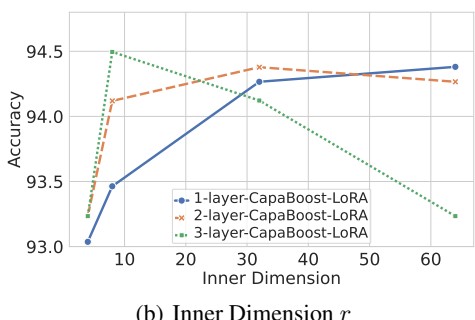

(a) Mask Density (b) Inner Dimension $r$

Figure 4: **Ablation study for components of CAPABOOST-LoRA on RoBERTa-base model. (a)** Average performance of CAPABOOST-LoRA with different masks, same mask, and Dropout over different sparsity on SST-2-10k dataset. We use two tied layers with a preset LoRA rank value of 8. **(b)** Average performance of CAPABOOST-LoRA under different rank values and number of tied layers when density= 0.5 on SST-2-10k dataset. Results are averaged over three trials.

Table 16: **Performance of random mask and N:M sparsity mask on the GLUE tasks.** This experiment takes RoBERTa-base as pre-trained model with inner dimension=8, and considers two masking strategies: fully random masks and N:M sparsity random pruning masks.

| | Average | CoLA | RTE | STS-B | MRPC | QNLI | SST-2 | MNLI | QQP |
|---|---|---|---|---|---|---|---|---|---|
| Random-masking CAPABOOST-LoRA | $85.49_{\pm0.3}$ | $63.20_{\pm0.6}$ | $76.77_{\pm2.2}$ | $90.50_{\pm0.2}$ | $89.54_{\pm0.7}$ | $92.15_{\pm0.1}$ | $94.72_{\pm0.2}$ | $86.86_{\pm0.1}$ | $90.19_{\pm0.0}$ |
| 2:4 Sparsity CAPABOOST-LoRA | $85.51_{\pm0.1}$ | $64.27_{\pm0.2}$ | $75.93_{\pm0.9}$ | $90.52_{\pm0.1}$ | $89.62_{\pm1.6}$ | $92.11_{\pm0.1}$ | $94.53_{\pm0.2}$ | $86.90_{\pm0.1}$ | $90.23_{\pm0.1}$ |

Table 17: **Detailed comparison results with regularization-based baselines.** This experiment takes RoBERTa-base as pre-trained model with inner dimension=8. The underlying PEFT mechanism for all baselines is *PAdapter*.

| | Average | CoLA | RTE | STS-B | MRPC | QNLI | SST-2 | MNLI | QQP |
|---|---|---|---|---|---|---|---|---|---|
| AdaMix | $85.53_{\pm0.4}$ | $63.19_{\pm1.1}$ | $78.22_{\pm0.4}$ | $\mathbf{90.51}_{\pm0.0}$ | $90.11_{\pm0.4}$ | $92.83_{\pm0.2}$ | $94.31_{\pm0.1}$ | $86.79_{\pm0.1}$ | $89.91_{\pm0.1}$ |
| Same-mask-CAPABOOST-PAdapter | $85.61_{\pm0.2}$ | $62.77_{\pm0.8}$ | $77.98_{\pm0.8}$ | $90.16_{\pm0.2}$ | $89.46_{\pm0.3}$ | $92.66_{\pm0.1}$ | $94.50_{\pm0.0}$ | $86.90_{\pm0.0}$ | $90.41_{\pm0.1}$ |
| Dropout-CAPABOOST-PAdapter | $85.06_{\pm0.2}$ | $59.92_{\pm0.7}$ | $77.98_{\pm1.5}$ | $90.33_{\pm0.2}$ | $89.13_{\pm0.3}$ | $92.66_{\pm0.1}$ | $\mathbf{94.82}_{\pm0.1}$ | $86.51_{\pm0.0}$ | $89.09_{\pm0.1}$ |
| CAPABOOST-PAdapter | $\mathbf{86.33}_{\pm0.2}$ | $\mathbf{65.25}_{\pm0.9}$ | $\mathbf{79.42}_{\pm1.3}$ | $\mathbf{90.51}_{\pm0.0}$ | $\mathbf{90.36}_{\pm0.8}$ | $\mathbf{92.90}_{\pm0.1}$ | $94.61_{\pm0.1}$ | $\mathbf{87.00}_{\pm0.1}$ | $\mathbf{90.61}_{\pm0.0}$ |

# I    COMPARISON TO REGULARIZATION-BASED BASELINES

One potential question arising from our CAPABOOST approach is whether the performance gain is primarily due to the implicit regularization effect induced by the random masking. We address this concern via comparing CAPABOOST with three new baselines: AdaMix (Wang et al., 2022c), Dropout (Hinton et al., 2012), and same masking. In particular, AdaMix proposes to randomly route training examples from a batch of inputs via stochastically chosed PEFT modules and adopt a consistency regularization loss. The performance improvements of Dropout and same masking compared to the underlying PEFT modules are purely coming from regularization effects. Experiment results are shown in Table 17. Note that we use *PAdapter* as the underlying PEFT mechanism for all baselines in this experiment.

# J    ADDITIONAL RELATED WORK

## J.1    PARAMETER-EFFICIENT FINE-TUNING (PEFT)

The first line of work in PEFT picks up a subset of the original model parameters to update, such as top layers (Donahue et al., 2014), specific model layers (Gheini et al., 2021), and internal modules (Zaken et al., 2022). These brute-force selection approaches are effective in reducing trainable parameters, but only leading to sub-optimal performance. Thus, various scoring functions are used to measure the importance of each parameter (Sung et al., 2021; Ansell et al., 2022; Guo et al., 2021). However, these scoring functions are usually task-specific and require additional computation.

Another line of work proposes to share the pre-trained network and insert task-specific trainable modules to steer the behavior of neural networks, which greatly reduces storage costs (Zhao et al., 2020). In particular, HAdapter proposed in Houlsby et al. (2019) places adapter modules after each feed-forward and attention layer. For better efficiency, PAdapter (Pfeiffer et al., 2021) suggests using adapters after FFN and LayerNorm modules (Ba et al., 2016). Inspired by textual prompting methods (Sun & Lai, 2020; Liu et al., 2019; Jiang et al., 2020; Shin et al., 2020), Prefix-Tuning (Li & Liang, 2021) prepends additional prefix vectors to the keys and values in the attention layer. He et al. (2021) systematically examine existing progress and further proposes a diverse array of Adapter variants by transferring design elements from Prefix-Tuning. To remove inference latency introduced by incremental parameters, Hu et al. (2022) use a similar bottleneck structure with low-rank constraint, in which learned weights can be merged into the pre-trained network. However, the upper bound of the rank of trainable parameter matrices in these methods is largely affected by the preset rank values, leading to constrained model capacity. Our work aims to break such capacity constraints and enhance parameter efficiency in fine-tuning.

## J.2    Low-rank Properties in Deep Neural Networks

In the over-parameterized regime, it has been demonstrated in many deep learning tasks that neural networks enjoy low-rank properties after training (Oymak et al., 2019). Inspired by this insight, some works (Jaderberg et al., 2014; Sainath et al., 2013; Khodak et al., 2020) propose to reinforce neural network training performance by explicitly injecting the low-rank constraints and achieve great success in CNNs. Similarly, LoRA (Hu et al., 2022) and a number of follow-up works (Dettmers et al., 2023; Zhang et al., 2022a; Chavan et al., 2023; Chen et al., 2023) borrow this idea and suggest applying low-rank updates to a frozen pre-trained network for fine-tuning on downstream tasks. While state-of-the-art model architectures like transformers have been shown to present a low-rank dimensionality and representations (Aghajanyan et al., 2021; Wang et al., 2020), Bhojanapalli et al. (2020) point out that the low-rank of key and query projections in the multi-head attention modules bottlenecks the performance of transformers. Experiments in Lialin et al. (2023) also demonstrate that transformers with low-rank updates perform significantly worse than full-rank baseline in the training stage.

## J.3    Weight-Tied Models

Weight-tied models, also known as weight-sharing or weight-tying models, are a type of parameter-efficient neural network architecture, in which the same set of weights is used across different layers or parts of the input (Dehghani et al., 2019; Dabre & Fujita, 2019; Xia et al., 2019; Lan et al., 2020; Li et al., 2021; Takase & Kiyono, 2021). This architecture, as the backbone of most implicit models, has been widely studied in recent years for various tasks (Wang et al., 2019; Liu et al., 2020; Yang et al., 2018; Lan et al., 2020; Takase & Kiyono, 2021; Zhang et al., 2020; Bender et al., 2020; Xie et al., 2021; Li et al., 2021). For instance, the Universal Transformer proposed in Dehghani et al. (2019) ties the parameters through one Transformer layer; such an idea was later employed in Dabre & Fujita (2019); Lan et al. (2020). In addition, Xia et al. (2019) introduce an encoder-decoder architecture that shares parameters between the encoder and decoder parts. Xiao et al. (2019) propose a method to share attention weights to speed up the computation of Transformers. Takase & Kiyono (2021) take this idea further and proposes three strategies to tie the parameters of different layers (with various combinations), instead of just sharing the parameters of one layer with all layers.

The aforementioned studies focus on proposing strategies to tie different layers and do not introduce sparse masks to a shared layer like ours. As a separate line of research, weight-sharing in Neural Architecture Search (NAS) (Zhang et al., 2020; Bender et al., 2020; Xie et al., 2021) reduces the computational overhead by sampling distinct neural architectures from a super net using sparse masks, where exponentially many architectures share weights in the same super-network and the costly training procedure is performed only once.

## J.4    Parameter Pruning for PEFT Methods

Pruning is a widely used strategy to improve the efficiency of neural networks by detecting and removing redundant parameters. To demonstrate the efficacy of parameter pruning, Frankle & Carbin (2018) propose the Lottery Ticket Hypothesis and show that one can always find a sparse sub-network in a dense model that, when trained in isolation, can achieve comparable performance to the original dense model. This conclusion also holds for randomly initialized dense models. There exist three directions of pruning, namely i) pruning at initialization, ii) dynamic pruning during training, and iii)

pruning after training. The latter two normally prune model weights relying on either importance-based or regularization-based criteria. The idea of pruning at initialization initially also relies on the magnitude-based metric (Frankle & Carbin, 2019); however, it is debated in several follow-up studies (Su et al., 2020; Frankle et al., 2021; Wang et al., 2022b; He et al., 2022b) that random masks—uniformly sampled and only fixes the pruning ratio per layer—are equally effective as the previous lottery ticket (Frankle & Carbin, 2019). Note that the research of model pruning primarily aims to develop improved pruning criteria or suggest enhanced optimization strategies/objective functions. To the best of our knowledge, our idea of introducing various random masks to a parallel weight-tied model is novel.

The most relevant work to our method might be Bai et al. (2022), which extends the codebook idea and learns masks on top of a fixed random weight vector to represent diverse dense layers. Specifically, masks are used to select values from a random vector (i.e., codebook) and thus form layers with distinct structures. Zeng et al. (2023) concurrently propose ProPETL, which also possesses a number of shared weights, on which different masks are learned to encode layer-specific and task-specific knowledge. Our work, in contrast to these approaches, learns shared weights with deterministic random binary masks, leading to stronger model capability.

Another scoring-based approach AdaLoRA (Zhang et al., 2022a) dynamically distributes the parameter budget among different layers by iteratively pruning singular values of incremental parameters in correspondence to the importance metric. In addition to extra computation, AdaLoRA also requires a higher budget of trainable parameters at the beginning of training, which does not apply to some low-resource scenarios.

