# OpenReview forum: "Increasing Model Capacity for Free: A Simple Strategy for Parameter Efficient Fine-tuning"
_ICLR.cc/2024/Conference — ICLR 2024 poster_

### Official Review · Reviewer_LzY5 · 2023-10-30

**Soundness:** 3 good
**Presentation:** 3 good
**Contribution:** 2 fair
**Rating:** 6
**Confidence:** 3

**Summary:**

The paper presents parallel smaller rank multiple low-rank metrices to approximate the frozen model's performance via low rank approximation, during the fine-tuning phase. To do this, the authors propose CAPABOOST, a sparse low-rank approximation with random sparse mask that remain frozen throughout the fine-tuning phase. The authors show results with CAPABOOST to perform better than LoRA, pre-fix tuning and adapter based approaches.

**Strengths:**

1. The results are impressive.

2. The paper is well written with related works being presented thoroughly to motivate and differentiate the proposed solution from the existing ones.

3. The baselines and comparison points cover multiple variants of low rank approximation of PEFT.

**Weaknesses:**

1. This sentence is not clear, "Furthermore, experiments conducted in Lialin et al. (2023) reveal that transformers with low-rank updates significantly underperform compared to full-rank baselines during pre-training.".. does it mean the Low rank adaptation during fine-tuning perform poorly? If so, how the LoRA or similar models perform close to baseline full-model fine-tuning? I suppose event the foundation of this paper, assumes that low rank adaptation is useful, and then tries to improve their rank. So, please clarify here.

2. Generation of mask randomly does increase storage cost, as we need to store the index location of the mask while performing forward and backward pass. So, it being storage free is not really a correct statement.

3. The authors claims of speeding up the sparse operation via Nvidia sparse tensor core, however, that is only applicable for N:M sparsity, which the current sparse mask generation process does not satisfy.

4. Can you explain with an example how with sparsity of 0.5 you can have more low rank matrices? And at what point it surpasses the params for a scenario when the sparsity is 1.0 (meaning that of Eq. 2)

5. Isn't capaboost with d>2 would require more fine-tuning params with s=0.5?

**Questions:**

1. When the author says training phase with the random generated frozen mask, does he/she mean fine-tuning phase?

2. The author should not claim as "for free", as with sparsity = 0.5 would often need storage that can be more than that with no sparsity. Also, additional tensor sparse operation  is needed making the latency to grow a little up, as it is not really N:M sparsity. So, please tone down on your claim, title, and writing.

---

> ### Author Response · Authors · 2023-11-17
> **Official Comment by Authors (1/4)**
>
> Dear reviewer LzY5:
>
> Thank you very much for the review and feedback, we kindly address your questions as follows.
>
> > 1. This sentence is not clear, "Furthermore, experiments conducted in Lialin et al. (2023) reveal that transformers with low-rank updates significantly underperform compared to full-rank baselines during pre-training.".. does it mean the Low rank adaptation during fine-tuning perform poorly? If so, how the LoRA or similar models perform close to baseline full-model fine-tuning? I suppose event the foundation of this paper, assumes that low rank adaptation is useful, and then tries to improve their rank. So, please clarify here.
> >
>
> First we apologize for the previous inaccurate expression and have since updated the paper.
>
> We do acknowledge and agree that low-rank adaptation is a useful and efficient fine-tuning method capable of reaching or even exceeding full-model fine-tuning performance in certain cases. However, it's important to note that it can underperform when the fine-tuning weight rank is insufficient. This is supported by Figure 4 in [a], Figure 2 in [b], and our Figure 4(b) and Figure 5. By increasing the rank with CapaBoost, we can enhance the performance of low-rank adaptation.
>
> > 2. Generation of mask randomly does increase storage cost, as we need to store the index location of the mask while performing forward and backward pass. So, it being storage free is not really a correct statement.
> >
>
> First we apologize for the misunderstanding we caused. We have revised our statement in the manuscript with the blue color.
>
> During fine-tuning stage, the pruning mask $m_1,m_2$ in CapaBoost are randomly generated and fixed, ideally, we only need to store random seed used to generate masks, rather than saving pruning mask index. Each time during forward and backward pass, we regenerate pruning mask using same random seed and discard masks after use, which may only induce temporary storage.
>
> Additionally, we would also like to emphasize that the dense fine-tuned weight matrix $W$ in PEFT methods can be partially pruned.
>
> Starting from Equation 3 in manuscript, we use an **example** to demonstrate this point. Suppose we have a 2-layer CapaBoost framework with 50\% sparsity, which can be written as $z=(W\odot m_1+W\odot m_2)X+b$,
>
>  we have $X=[x_1,x_2,x_3,x_4]^{T}$, $W=\begin{bmatrix}w_{11}& w_{12}& w_{13}& w_{14} \\\ w_{21}& w_{22}& w_{23}& w_{24} \\\ w_{31}& w_{32}& w_{33}& w_{34} \\\ w_{41}& w_{42}& w_{43}& w_{44}\\\ \end{bmatrix}$, $m_1=\begin{bmatrix}1& 0& 1& 0 \\\ 1& 1& 0& 0  \\\ 0& 1& 1& 0 \\\ 0& 0& 1& 1 \\\ \end{bmatrix}$, $m_2=\begin{bmatrix}1& 1& 0& 0 \\\ 1& 0& 0& 1  \\\ 1& 0& 0& 1 \\\ 0& 0& 1& 1 \\\ \end{bmatrix}$
>
> since elements $w_{14}, w_{23}, w_{41}, w_{42}$ are  pruned in both mask $m_1$ and $m_2$, they are never used in computation. As a result, they can also be pruned in fine-tuned matrix $W$. Therefore, CapaBoost always leads to less trainable parameters. The accurate parameter reduction number can refer to answer for question 5.
>
> Given the reduction in number of trainable parameters and negligible increase in temporary mask storage, our method in general doesn’t require more storage compared to original PEFT methods without CapaBoost.
>
> [a] He, Junxian, et al. "Towards a Unified View of Parameter-Efficient Transfer Learning." International Conference on Learning Representations. 2021.
>
> [b] Qingru Zhang, Minshuo Chen, Alexander Bukharin, Pengcheng He, Yu Cheng, Weizhu Chen, and Tuo
> Zhao. Adaptive budget allocation for parameter-efficient fine-tuning. In The Eleventh International
> Conference on Learning Representations, 2022.

---

> ### Author Response · Authors · 2023-11-18
> **Official Comment by Authors (2/4)**
>
> > 3. The authors claims of speeding up the sparse operation via Nvidia sparse tensor core, however, that is only applicable for N:M sparsity, which the current sparse mask generation process does not satisfy.
> >
>
> Thank you so much for pointing out this problem. We now kindly include the GLUE test results with RoBERTa-base as the pre-trained model and inner dimension = 8, and consider two masking strategies: fully random masks and N:M sparsity random pruning masks. These results are detailed in the table below.
>
> The results indicate that CapaBoost performs similarly with both fully random masks and N:M sparsity random masks. Thus, we can directly replace fully random masks with N:M sparsity random masks and benefit from the hardware acceleration provided by the Nvidia sparse tensor core. In our experiment, the fine-tuned weight matrix  $W$ with N:M sparsity displays the same rank as $W$ with fully random masks. Therefore, it’s not surprising that the result is similar.
>
> |  | Average | CoLA | RTE | STSB | MRPC | QNLI | SST2 | MNLI | QQP |
> | --- | --- | --- | --- | --- | --- | --- | --- | --- | --- |
> | Random-masking Capaboost LoRA | $85.49 \pm 0.3$ | $63.20 \pm 0.6$ | $76.77 \pm 2.2$ | $90.50 \pm 0.2$ | $89.54 \pm 0.7$ | $92.15 \pm 0.1$ | $94.72 \pm 0.2$ | $86.86 \pm 0.1$ | $90.19 \pm 0.0$ |
> | 2:4 Sparsity Capaboost LoRA | $85.51 \pm 0.1$ | $64.27 \pm 0.2$ | $75.93 \pm 0.9$ | $90.52 \pm 0.1$ | $89.62 \pm 1.6$ | $92.11 \pm 0.1$ | $94.53 \pm 0.2$ | $86.90 \pm 0.1$ | $90.23 \pm 0.1$ |
>
> > 4. Can you explain with an example how with sparsity of 0.5 you can have more low rank matrices? And at what point it surpasses the params for a scenario when the sparsity is 1.0 (meaning that of Eq. 2)
> >
>
> Thank you for the question.
>
> In the manuscript's Equation 2 (not CapaBoost), since $W_1,W_2...W_d$ are all distinct and don’t derive from a common matrix, the total parameter number is $d$ times than parameter number of $W$.
>
> However, in CapaBoost case, as explained in the 2-layer CapaBoost example of question 2, with two different random masks $m_1,m_2$ applying to the same $W$, we can obtain two different weight matrices  $W\odot m_1, W\odot m_2$. Similarly, in the $d$ layer of CapaBoost, with $d$ different random masks, we can obtain $d$ diverse sparse weight matrices, where all matrices elements are a subset of dense matrix $W$. The parameter number in CapaBoost will never exceed $W$ in the original $z=WX+b$ since all sparse weight matrices originate from $W$. Thus, our maximum parameter number is the same as $W$.
>
> > 5. Isn't CapaBoost with d>2 would require more fine-tuning params with s=0.5?
> >
>
> Thank you for the question. Suppose the original matrix $W$ has a parameter number $n$, for the $d$-layer CapaBoost with sparsity of $s$, if all pruning masks are randomly generated, the parameter number is $(1-s^d)*n$ according to the binomial distribution. If $s=0.5, d=2$, we only need 75\% of original parameter number. With more layers, CapaBoost will require more fine-tuning parameters, but it will never exceed $n$.
>
> > When the author says training phase with the random generated frozen mask, does he/she mean fine-tuning phase?
> >
>
> Yes, training phase refers to fine-tuning on the pre-trained model.

---

> ### Author Response · Authors · 2023-11-18
> **Official Comment by Authors (3/4)**
>
> > The author should not claim as "for free", as with sparsity = 0.5 would often need storage that can be more than that with no sparsity. Also, additional tensor sparse operation is needed making the latency to grow a little up, as it is not really N:M sparsity. So, please tone down on your claim, title, and writing.
> >
>
> Thank you so much for your comment. We apologize for our previous unclear expression, and we would like to further explain as follows.
>
> 1. Inference phase. Based on Equation 4 and 5 in manuscripts, we can combine parallel weights into one and get $B^* = B\odot m_{b_1}+...+ B\odot m_{b_d}$ , and $A^* = A\odot m_{a_1}+...+A\odot m_{a_d}$, where both $B^*$ and $A^*$ have the same size as corresponding original PEFT methods. Thus, CapaBoost inference phase is ‘for free’, as there is no extra storage or computational cost.
> 2. Fine-tuning phase. Our CapaBoost implementation is for free because of the fact that
>     - As illustrated in response to question 2 and question 5, our parameter number is less than corresponding PEFT methods without CapaBoost. Mask storage can also be avoided through same random seed regeneration, and thus the storage cost is smaller than original PEFT methods.
>     - As shown in Figure 4 and Figure 5, a 2-layer CapaBoost with sparsity=0.5 is enough to achieve competitive performance, which is also the setting we used in most experiments presented in the paper(Table 2,3,4 and 5). With this setting, the theoretical FLOPs of CapaBoost are as same as original PEFT methods. With our newly implemented N:M sparsity random masks, we can enjoy Nvidia sparse tensor core acceleration and realize comparable FLOPs compared to original PEFT methods in reality. Therefore, our CapaBoost implementation in experiments is also nearly for free.
>     - We also acknowledge that CapaBoost is a general framework, its FLOPs depends on selection of layer number and mask sparsity, and might be more or less than corresponding PEFT methods without CapaBoost. However, our implementation in the manuscript’s experiments is nearly for free.

---

> ### Author Response · Authors · 2023-11-21
> **Official Comment by Authors (4/4)**
>
> Dear Reviewer LzY5,
>
> Thank you again for your review and comments. Since the discussion period is closing soon, we are wondering whether our responses have addressed your concerns. If not, we would be happy to continue the discussion and provide further clarifications or explanations.

---

### Official Review · Reviewer_7oiV · 2023-10-31

**Soundness:** 2 fair
**Presentation:** 3 good
**Contribution:** 2 fair
**Rating:** 8
**Confidence:** 3

**Summary:**

This paper provides an approach to improving the performance of PEFT methods. They propose to "expand" the effective rank of the learned  low-rank matrices by generating different masks for masking the same base low-rank matrix.
They show through ablations and experiments that their approach is complementary with other PEFT methods.

**Strengths:**

In general, I think the method is a clever idea. I believe this idea of generating "pseudo-independent" parallel matrices whilst saving storage by applying masks to a base matrix is novel. In the presence of specialized hardware, the sparse matrix form and the use of random-seed to mark masks could lead to considerable memory and compute savings.

**Weaknesses:**

## Weaknesses
* Flop improvement from method only exist in the presence of specialized hardware for sparse matrices.
* Were any of the non-capaboost methods regularized ?  One possible explanation for the improvements of capaboost -- is that the random masking creates implicit regularization. It would be good to compare the performance to other methods, once they also enjoy some level of regularization.
* I find it hard to understand / believe Table 1 for multiple reasons. First, the claim is that you generally get a d - 1 increase over LoRA (remark 3.3) but  you consistently report dx instead (d - 1)x increase. Second, the basis for the intuition for the dx expansion, is theorem 3.1 which assumes that X and Y are generated independently from Gaussians. In general, this would be the upper bound on the rank of the sum. The matrices you instantiate are not independent (since they are generated from the same B,A combination -- though masked) ... but somehow your results (from Table 1) suggest that you are always achieving this upper bound which is quite surprising. And what about the case where the rank of A, B = 1 ?

There seem to be quite a few inaccurate statements peppered through the paper :
1. "Fine-tuning large pre-trained foundation models, such as the 175B GPT-3, has become the prevailing approach for downstream tasks." -- I don't think this is true -- the defacto approach for using models at these scales is prompting and not fine-tuning.

2. "..... replacing a single layer’s heavyweight matrix w ∈ Rd1×d2 with two consecutive layers possessing much smaller inner dimension r, such as B ∈ Rd1 ×r , A ∈ Rr×d2 . This approach results in a significant reduction in parameter count (evident in Adapters and LoRA), achieving performance comparable to full-size fine-tuning while retaining only 1% of trainable parameters" -- the original matrices are not "replaced" by these smaller ones but this statement makes it seem like the full parameter versions are swapped for the low-rank version


In general, I am not sure I am convinced that the method works for the reasons mention in the paper. I am skeptical of the proposed rank expansion framing -- and I am more convinced that there might be some effective regularization occurring due to the random masks.

Note : I am willing to raise my score if my concerns / questions  are addressed.

**Questions:**

1. How do you measure/compute the flops in your paper ? You only mention that its "relative to lora"
2. I am surprised that full finetuning from Figure 4b performs substantially worse that the low rank methods. What hyper-parameters were used ? Was there regularization when training this model ?

---

> ### Author Response · Authors · 2023-11-17
> **Official Comment by Authors (1/3)**
>
> Dear reviewer 7oiV:
>
> Thank you very much for the review and feedback. We kindly address your questions as follows:
>
> > Flop improvement from method only exist in the presence of specialized hardware for sparse matrices.
> >
>
> Thank you for bringing out this question. We fine-tuned the RoBERTa-base model on the GLUE dataset with an inner dimension of 8, and considered two masking strategies: fully random masks and N:M sparsity random pruning masks. We present experimental results in the table below. CapaBoost with N:M sparsity shows comparable performance to fully random pruning masks. These N:M sparsity masks enable us to have Nvidia sparse tensor core (e.g., A100) acceleration and achieve FLOPs improvement in the real world.
>
> |  | Average | CoLA | RTE | STSB | MRPC | QNLI | SST2 | MNLI | QQP |
> | --- | --- | --- | --- | --- | --- | --- | --- | --- | --- |
> | Random-masking Capaboost LoRA | $85.49 \pm 0.3$ | $63.20 \pm 0.6$ | $76.77 \pm 2.2$ | $90.50 \pm 0.2$ | $89.54 \pm 0.7$ | $92.15 \pm 0.1$ | $94.72 \pm 0.2$ | $86.86 \pm 0.1$ | $90.19 \pm 0.0$ |
> | 2:4 Sparsity Capaboost LoRA | $85.51 \pm 0.1$ | $64.27 \pm 0.2$ | $75.93 \pm 0.9$ | $90.52 \pm 0.1$ | $89.62 \pm 1.6$ | $92.11 \pm 0.1$ | $94.53 \pm 0.2$ | $86.90 \pm 0.1$ | $90.23 \pm 0.1$ |
>
> > Were any of the non-CapaBoost methods regularized? One possible explanation for the improvements of CapaBoost -- is that the random masking creates implicit regularization. It would be good to compare the performance to other methods, once they also enjoy some level of regularization.
> >
>
> Thank you so much for your question. To address this concern, we first include two other variants of CapaBoost: 1) same-mask CapaBoost,  2) Dropout CapaBoost and 3) AdaMix [a] which has explicit regularization.
>
> - In same-mask CapaBoost, we maintain CapaBoost parallel structure and replace the different parallel pruning masks with one identical mask. With this design, same-mask CapaBoost still benefits from implicit regularization but no longer possesses the rank expansion property.
> - Similarly, we replace different parallel masks by Dropout in Dropout CapaBoost.
> - Explicit regularized method AdaMix [a].
>
> We reported the GLUE test results with RoBERTa-base as the pre-trained model and an inner dimension of 8 in the table below. The results show that CapaBoost can both outperform explicit regularized and implicit regularized competitors. It’s also worth noticing that all non-CapaBoost methods in our paper use regularization method with control of weight decay.
>
> |  | Average | CoLA | RTE | STSB | MRPC | QNLI | SST2 | MNLI | QQP |
> | --- | --- | --- | --- | --- | --- | --- | --- | --- | --- |
> | AdaMix | $85.53 \pm 0.4$ | $63.19 \pm 1.1$ | $78.22 \pm 0.4$ | $\mathbf{90.51} \pm 0.0$ | $90.11 \pm 0.4$ | $92.83 \pm 0.2$ | $94.31 \pm 0.1$ | $86.79 \pm 0.1$ | $89.91 \pm 0.1$ |
> | CapaBoost-PAdapter (same mask) | $85.61 \pm 0.2$ | $62.77 \pm 0.8$ | $77.98 \pm 0.8$ | $90.16 \pm 0.2$ | $89.46 \pm 0.3$ | $92.66 \pm 0.1$ | $94.50 \pm 0.0$ | $86.90 \pm 0.0$ | $90.41 \pm 0.1$ |
> | CapaBoost-PAdapter (Dropout) | $85.06 \pm 0.2$ | $59.92 \pm 0.7$ | $77.98 \pm 1.5$ | $90.33 \pm 0.2$ | $89.13 \pm 0.3$ | $92.66 \pm 0.1$ | $\mathbf{94.82} \pm 0.1$ | $86.51 \pm 0.0$ | $89.09 \pm 0.1$ |
> | CapaBoost-PAdapter | $\mathbf{86.33} \pm 0.2$ | $\mathbf{65.25} \pm 0.9$ | $\mathbf{79.42} \pm 1.3$ | $\mathbf{90.51} \pm 0.0$ | $\mathbf{90.36} \pm 0.8$ | $\mathbf{92.90} \pm 0.1$ | $94.61 \pm 0.1$ | $\mathbf{87.00} \pm 0.1$ | $\mathbf{90.61} \pm 0.0$ |
>
> Based on the above observation, we agree regularization may also contribute to CapaBoost’s superior performance, but most of its improvement stems from rank expansion.
>
> Remarks: 1) Some ablation study about same-mask and Dropout are also shown in Figure.4 in the paper 2) The experiment here has a different inner dimension from Table.2 in the manuscript.
>
> [a] Wang, Yaqing, et al. "AdaMix: Mixture-of-Adaptations for Parameter-efficient Model Tuning." Proceedings of the 2022 Conference on Empirical Methods in Natural Language Processing. 2022.

---

> ### Author Response · Authors · 2023-11-17
> **Official Comment by Authors (2/3)**
>
> > I find it hard to understand / believe Table 1 for multiple reasons. First, the claim is that you generally get a d - 1 increase over LoRA (remark 3.3) but you consistently report dx instead (d - 1)x increase. Second, the basis for the intuition for the dx expansion, is theorem 3.1 which assumes that X and Y are generated independently from Gaussians. In general, this would be the upper bound on the rank of the sum. The matrices you instantiate are not independent (since they are generated from the same B,A combination -- though masked) ... but somehow your results (from Table 1) suggest that you are always achieving this upper bound which is quite surprising. And what about the case where the rank of A, B = 1 ?
> >
>
> Thank you for your question. We apologize for any confusion caused by our previous explanation. In the paper we intended to express CapaBoost’s rank is d times of LoRA’s rank (d-1 times larger), we have revised the manuscript with blue color to clarify this point.
>
> To better address your concern, we conducted a simple Monte Carlo simulation of a 2-layer CapaBoost with rank of $B$ and $A$ $=1,2,8,16,32$. In this simulation, we randomly generate $B$, $A$, their corresponding pruning masks $m_{A1},m_{A2}, m_{B1}, m_{B2}$ and compare rank of $BA$ and $(B\odot m_{B1})(A\odot m_{A1})+(B\odot m_{B2})(A\odot m_{A2})$. The results from 10,000 simulations consistently show that $(B\odot m_{B1})(A\odot m_{A1})+(B\odot m_{B2})(A\odot m_{A2})$ always achieves 2 times rank expansion compared to $BA$, which clearly indicates this rank expansion is not a coincidence. One possible explanation behind is that with different random pruning masks, $(B\odot m_{B1})(A\odot m_{A1})$ and $(B\odot m_{B2})(A\odot m_{A2})$ are of great difference and can be viewed as pseudo-independent. But since the $B,A$‘s distribution after training is unknown, it's challenging to provide a definitive proof.
>
> > "Fine-tuning large pre-trained foundation models, such as the 175B GPT-3, has become the prevailing approach for downstream tasks." -- I don't think this is true -- the defacto approach for using models at these scales is prompting and not fine-tuning.
> >
>
> > "..... replacing a single layer’s heavyweight matrix w ∈ Rd1×d2 with two consecutive layers possessing much smaller inner dimension r, such as B ∈ Rd1 ×r , A ∈ Rr×d2 . This approach results in a significant reduction in parameter count (evident in Adapters and LoRA), achieving performance comparable to full-size fine-tuning while retaining only 1\% of trainable parameters" -- the original matrices are not "replaced" by these smaller ones but this statement makes it seem like the full parameter versions are swapped for the low-rank version
> >
>
> Thank you so much for pointing it out, we have updated our expression in PDF.
>
> > How do you measure/compute the flops in your paper ? You only mention that its "relative to lora"
> >
>
> Thank you for the question. We calculate FLOPs needed in each layer based on layer type, size and Sparse core acceleration, and then calculate total number. For convenience, we only show the FLOPs factor relative to that of LoRA
>
> > I am surprised that full finetuning from Figure 4b performs substantially worse that the low rank methods. What hyper-parameters were used ? Was there regularization when training this model ?
> >
>
> Thank you for the question you bring out. Figure 4b is an experiment on CoLA dataset. The full fine-tuning hyper-parameters settings follow [b], which is regularized via using weight decay.
>
> The results remain consistent with those in Table 2 and Table 3 in our paper, where full fine-tuning consistently shows significantly worse performance than CapaBoost in CoLA dataset. Similar findings are also reported in [b] and [c]. This may because full fine-tuning contains large number of parameters and tends to overfit CoLA dataset, even with help of regularization.
>
> In the next step, we plan to cover this experiment on the entire GLUE dataset to make the result more convincing.
>
> [b]Guangtao Zeng, Peiyuan Zhang, and Wei Lu. One network, many masks: Towards more parameterefficient transfer learning. In Proceedings of ACL, 2023.
>
> [c]Qingru Zhang, Minshuo Chen, Alexander Bukharin, Pengcheng He, Yu Cheng, Weizhu Chen, and Tuo Zhao. Adaptive budget allocation for parameter-efficient fine-tuning. In The Eleventh International Conference on Learning Representations, 2022a.

---

> ### Author Response · Authors · 2023-11-21
> **Official Comment by Authors (3/3)**
>
> Dear Reviewer 7oiV,
>
> Thank you again for your review and comments. Since the discussion period is closing soon, we are wondering whether our responses have addressed your concerns. If not, we would be happy to continue the discussion and provide further clarifications or explanations.

---

> > ### Comment · Reviewer_7oiV · 2023-11-22
> > **Thanks for response and 1 more clarification**
> >
> > Thanks for clarifying some of the questions I posed.
> > I thought I about it some more and I think rank expansion might indeed be what's happening.
> > Specifically, I think having a simple example in the paper like below :
> > - say your rank 1 matrix is [1, 1]
> > - say your binary masks are  [1, 0], [0, 1]
> > - then by taking the Hadamard product of the 2 binary masks with [1, 1], we get back   [1, 0], [0, 1], which span R^2, which can be seen as "rank" expansion.
> >
> > I would encourage the authors to think about a very simple example -- like the one  I give above (but obviously w.r.t matrix products as in the PEFT methods) and include that in the paper to make it more obvious that rank expansion is not "magic".
> >
> > I have updated my score :)

---

> > > ### Author Response · Authors · 2023-11-22
> > > **Final Response to Reviewer 7oiV**
> > >
> > > Dear Reviewer 7oiV,
> > >
> > > We would like to express our heartfelt appreciation for your valuable feedback and suggestions. We will incorporate a toy example as you suggested into the next revision to further make our idea more motivated.

---

### Official Review · Reviewer_ZJAd · 2023-10-31

**Soundness:** 3 good
**Presentation:** 3 good
**Contribution:** 3 good
**Rating:** 6
**Confidence:** 3

**Summary:**

In this paper, CAPABOOST is introduced as a versatile framework with the aim of amplifying model capacity in a range of Parameter Efficient Fine-Tuning (PEFT) methods, including those like Adapters and LoRA. This is achieved by integrating techniques such as model pruning and weight sharing. CAPABOOST's primary objective is to expand model capacity without adding extra computational overhead, leading to enhanced performance compared to conventional PEFT approaches. The paper provides empirical evidence of CAPABOOST's effectiveness by highlighting significant performance enhancements, reduced parameter counts, and equivalent or reduced floating-point operations (FLOPs).

**Strengths:**

1) The central concern addressed in this paper pertains to a key issue in existing PEFT methods: while they effectively reduce the parameter count, they often impose limitations on the model's rank.
2) Through extensive empirical results, it is evident that CAPABOOST surpasses current benchmarks by achieving substantial reductions in parameter count while preserving the same or fewer FLOPs.
3) The authors commit to providing both the code and benchmarks, which holds the potential to enhance the reproducibility of their research.

**Weaknesses:**

Regarding the computer vision (CV) task experiments, the paper only presents results for LoRA and LoRA+CAPABOOST. It would be advantageous to incorporate the results of other baseline methods, such as ProPETL[1], to provide a more comprehensive comparison.

[1] One network, many masks: Towards more parameter efficient transfer learning

**Questions:**

Kindly refer to the weaknesses.

---

> ### Author Response · Authors · 2023-11-18
> **Official Comment by Authors (1/3)**
>
> Dear reviewer ZjAd:
>
> Thank you so much for your positive rating and constructive advice. The experiment of ProPETL for Vtab-1k task is running; we will update it to the paper as soon as possible.

---

> ### Author Response · Authors · 2023-11-20
> **Official Comment by Authors (2/3)**
>
> Dear reviewer ZjAd:
>
> Thank you for your patience. We kindly include additional benchmarks in the Vtab-1k experiment and present you with the new results in the table below.
>
> 1. Visual Prompt Tuning (VPT) [a]. VPT prepends a set of learnable tokens to the input of a Transformer block, which can be viewed as adding some learnable pixels in the input space.
> 2. NOAH [b]: NOAH is a neural architecture search method (NAS) that learns the optimal design of several parameter-efficient subnetworks, including VPT, LoRA and Adapter. This type of neural search method requires more parameter numbers and computational costs.
> 3. ProPETL[c]. Since there is no existing implementation for the ProPETL method on Vtab-1k tasks, we reproduced it based on open source code provided by the author and fine-tuned it for optimal results. The results indicate that ProPETL-LoRA offers a performance gain of about 0.2% compared to the original LoRA. This is consistent with the slight performance gain ProPETL-LoRA has in the GLUE dataset.
>
> The results indicate that CapaBoost-LoRA still outperforms all other competitors, demonstrating the effectiveness of the CapaBoost framework.
>
> |  |  |  |  |  |  |  |  |  |  |  |  |  |  |
> | --- | --- | --- | --- | --- | --- | --- | --- | --- | --- | --- | --- | --- | --- |
> | Model Name | Cifar-100 | DTD | Flower102 | Pets | SVHN | Eurosat | Resisc45 | Clevr-Count | Clevr-Dist | DMLab | dSpr-Ori | sNORB-Azim | Average |
> | VPT-Deep* | $\mathbf{78.80}$ | $65.80$ | $98.00$ | $88.30$ | $78.10$ | $\mathbf{96.10}$ | $83.40$ | $68.50$ | $60.00$ | $46.50$ | $47.90$ | $\mathbf{32.90}$ | $70.36$ |
> | NOAH* | $69.60$ | $70.20$ | $\mathbf{99.10}$ | $90.40$ | $86.10$ | $95.40$ | $83.90$ | $\mathbf{82.80}$ | $68.90$ | $49.90$ | $\mathbf{48.30}$ | $32.80$ | $73.11$ |
> | LoRA | $67.10 \pm 0.3 $ | $69.89 \pm 0.5 $ | $98.95 \pm 0.1 $ | $90.58 \pm 0.3 $ | $84.20 \pm 0.7 $ | $95.34 \pm 0.2 $ | $85.93 \pm 0.1 $ | $82.68 \pm 0.3 $ | $67.92 \pm 0.5 $ | $49.68 \pm 0.2 $ | $45.54 \pm 0.2 $ | $30.80 \pm 0.4$ | $72.38$ |
> | ProPETL-LoRA | $69.50 \pm  0.4$ | $70.48 \pm 0.5$ | $98.91 \pm 0.0$ | $90.84 \pm 0.2$ | $85.64 \pm 0.9$ | $95.41\pm 0.3$ | $85.35\pm 0.1$ | $81.99\pm 0.2$ | $67.26\pm 0.4$ | $49.79\pm 0.7$ | $46.97\pm 0.5$ | $28.74\pm 0.3$ | $72.57$ |
> | CapaBoost-LoRA (d=2) | $69.28 \pm 0.5 $ | $71.45 \pm 0.3 $ | $99.06 \pm 0.1 $ | $\mathbf{91.21} \pm 0.3 $ | $85.01 \pm 1.1 $ | $95.55 \pm 0.2 $ | $\mathbf{86.48} \pm 0.1 $ | $\mathbf{82.80} \pm 0.2 $ | $69.05 \pm 0.4 $ | $50.58 \pm 0.8 $ | $46.04 \pm 0.1 $ | $31.94 \pm 0.2$ | $73.20$ |
> | CapaBoost-LoRA (d=4) | $69.01 \pm 0.1 $ | $\mathbf{72.18} \pm 0.6 $ | $99.06 \pm 0.0 $ | $91.20 \pm 0.2 $ | $\mathbf{87.07} \pm 0.7 $ | $95.76 \pm 0.1 $ | $85.96 \pm 0.2 $ | $82.77 \pm 0.3 $ | $\mathbf{69.11} \pm 0.3 $ | $\mathbf{50.90} \pm 0.9 $ | $45.84 \pm 0.3 $ | $32.09 \pm 0.3$ | $\mathbf{73.41}$ |
>
> Note: * indicates the results are reported in prior works [a] and [b] respectively.
>
> Thank you again for your feedback. We respectfully request that you reconsider the review score if we have addressed all of your concerns.
>
> [a] Jia, Menglin, et al. "Visual Prompt Tuning." *European Conference on Computer Vision*. 2022.
>
> [b] Zhang, Y., Zhou, K., & Liu, Z. (2022). Neural Prompt Search. *ArXiv, abs/2206.04673*.
>
> [c] Guangtao Zeng, Peiyuan Zhang, and Wei Lu. One network, many masks: Towards more parameter-efficient transfer learning. In Proceedings of ACL, 2023.

---

> ### Author Response · Authors · 2023-11-21
> **Official Comment by Authors (3/3)**
>
> Dear Reviewer ZJAd,
>
> Thank you again for your review and comments. Since the discussion period is closing soon, we are wondering whether our responses have addressed your concerns. If not, we would be happy to continue the discussion and provide further clarifications or explanations.

---

### Meta-Review · Area_Chair_GnGm · 2023-12-04

**Metareview:**

This paper proposes an approach for cheaply increasing the capacity of parameter-efficient finetuning (PEFT) methods, in particular LoRA and Adapters. This is done via having randomly-generated masks applied to the shared PEFT components, and then running the input through these "masked" PEFT layers whose outputs are aggregated. (The random masks remain fixed from the start and thus can be represented through a seed, thus obviating the need to store these masks separately).

This is a nice, simple idea that results in nontrivial empirical improvements. On the downside, while there is a decrease in FLOPs, I suspect that there will still be some overhead in inference because in practice the sparse operations on GPU are not actually faster (especially at the ~50% sparsity rates that are explored in this paper).

**Justification For Why Not Higher Score:**

While the idea is nice, practical implementations will not result in any wall clock savings.

**Justification For Why Not Lower Score:**

The simplicity of the idea combined with nontrivial gains over baselines.

---

### Decision · Program_Chairs · 2024-01-16

Accept (poster)